# Spatially resolved photoluminescence analysis of the role of Se in CdSe$_x$Te$_{1-x}$ thin films

A. R. Bowman [1,2,3,5], J. F. Leaver [4,5], K. Frohna [2,3], S. D. Stranks [2,3], G. Tagliabue [1] ✉ & J. D. Major [4] ✉

Evidence from cross-sectional electron microscopy has previously shown that Se passivates defects in CdSe$_x$Te$_{1-x}$ solar cells, and that this is the reason for better lifetimes and voltages in these devices. Here, we utilise spatially resolved photoluminescence measurements of CdSe$_x$Te$_{1-x}$ thin films on glass to directly study the effects of Se on carrier recombination in the material, isolated from the impact of conductive interfaces and without the need to prepare cross-sections through the samples. We find further evidence to support Se passivation of grain boundaries, but also identify an increase in below-bandgap photoluminescence that indicates the presence of Se-enhanced defects in grain interiors. Our results show that whilst Se treatment, in tandem with Cl passivation, does increase radiative efficiencies in CdSe$_x$Te$_{1-x}$, it simultaneously increases the defect content within the grain interiors. This suggests that although it is beneficial overall, Se incorporation will still limit the maximum attainable optoelectronic properties of CdSe$_x$Te$_{1-x}$ thin films.

CdTe-based solar cells are currently the only commercial competitor to silicon cells and offer the lowest levelized cost of electricity (LCOE) of any photovoltaic technology for utility-scale generation[1]. However, the record power conversion efficiency for a small-area CdTe cell is 22.3%, markedly lower than the 26.8% for crystalline silicon[2] despite the higher theoretical maximum efficiency for CdTe[3]. As such, there is significant scope to improve CdTe device efficiencies and further drive down the LCOE of solar electricity generation.

A relatively recent innovation in CdTe-based solar cells has been the introduction of a Se-alloyed CdSe$_x$Te$_{1-x}$ region at the light-facing side of the absorber, with a graded composition from higher Se content at the light-facing side towards pure CdTe at the rear of the device. The CdSe$_x$Te$_{1-x}$ region has a lower bandgap than CdTe for moderate Se content[4,5], resulting in a bandgap grading in these devices (from lower at the light-facing side to higher at the rear contact). The lower bandgap at

the light-facing side of CdSe$_x$Te$_{1-x}$ device extends light absorption further into the infrared, increasing the photocurrent compared to pure CdTe devices[6,7]. Surprisingly, CdSe$_x$Te$_{1-x}$ devices have been demonstrated with similar open circuit voltages ($V_{OC}$) to pure CdTe devices[8,9], indicating a reduction in $V_{OC}$ losses that compensate for the lower bandgap. Cathodoluminescence (CL) measurements have shown increased luminescence, especially at grain boundaries, associated with higher Se content[10–12], suggesting that Se passivates defects in CdSe$_x$Te$_{1-x}$ devices, leading to reduced $V_{OC}$ losses. However, cross-sectional CL measurements can be influenced by surface effects as a result of the need to create an exposed cross-section through the device via mechanical or ion beam polishing, as well as generating charges in significantly different intensities and regions of the device compared to optical illumination. Photoluminescence (PL) thus offers important complementary evidence of carrier recombination in intact material.

[1]Laboratory of Nanoscience for Energy Technologies (LNET), STI, École Polytechnique Fédérale de Lausanne (EPFL), Lausanne, Switzerland. [2]Cavendish Laboratory, Department of Physics, University of Cambridge, J.J. Thomson Avenue, Cambridge, UK. [3]Department of Chemical Engineering & Biotechnology, University of Cambridge, Philippa Fawcett Drive, Cambridge, UK. [4]Department of Physics and Stephenson Institute for Renewable Energy, University of Liverpool, Liverpool, UK. [5]These authors contributed equally: A. R. Bowman, J. F. Leaver. ✉e-mail: giulia.tagliabue@epfl.ch; jon.major@liverpool.ac.uk

Bulk photoluminescence measurements were recently employed on CdSe$_x$Te$_{1-x}$ devices to explore the role of Se[13]. Specifically, temperature-dependent PL suggested that, for some fabrication processes, samples containing Se had sub-bandgap defects that limited the maximum achievable $V_{OC}$. Similar sub-bandgap PL defects have also been observed by Hu et al.[14] in Se-containing samples at low temperatures. In both cases the nature of this defect and its relationship to sample morphology were not discussed. Microscale PL is a technique that can begin to answer these questions. It has previously been used to assess charge transport and passivation in both CdTe and CdSe$_x$Te$_{1-x}$ samples separately[15,16], and to explore the relationship of defects and grain boundaries to sample morphology in pure CdTe films[17] and CdSe$_x$Te$_{1-x}$ in giant grain samples[18]. Similarly, time-resolved and spatially-resolved PL measurements have revealed heterogeneities in charge lifetimes in CdSe-based solar cells[19], and two-photon PL microscopy has also been helpful in disentangling surface and bulk transport and recombination effects in CdTe samples[20,21], including passivation effects of dopants[22,23], and in CdSe$_x$Te$_{1-x}$ samples[24]. Notably, prior microscale PL studies have measured samples in full or partial device stacks. This presents a fundamental issue: it is impossible to disentangle spectroscopically whether changes in charge lifetime or luminescence strength originate from charge extraction or sample passivation processes (Supplementary Information Note 1). Furthermore, no study has investigated the role of Se in samples in a controlled manner with CdSe concentration being carefully varied between samples. Finally, to date, nearly all studies have focused on relative values, rather than absolute quantum efficiencies. Absorption – the key complementary measurement to PL – has not been recorded at the microscale.

Here we fabricated CdSe$_x$Te$_{1-x}$ thin films on glass by interdiffusion of separately deposited CdSe and CdTe layers, with varying thicknesses of CdSe from 0 nm to 200 nm to alter the Se content in the final film. We are able to assess the role of Se on luminescence and recombination by measuring the spatially resolved absorption and photoluminescence of the samples. Fabrication on glass also allows us to directly assess the impact on the absorber layer, separated from the remainder of the device structure (we discuss this further in Supplementary Information Note 1). Our results show that Se is distributed throughout the films and clearly passivates grain boundaries, in line with previous cross-sectional electron microscopy studies[10,25]. While Se alloying is shown not to significantly impact on the disorder of the main luminescence peak (as found from Urbach tails), we find it does increase the strength of below-bandgap defect luminescence. Our results suggest that these defect states are more prevalent in grain interiors, implying a distinction in Se behaviour between grain boundary and bulk. This work provides further support for grain boundary passivation by Se, but crucially also indicates that the effects of Se might not be wholly beneficial, introducing defect states which could provide an ultimate cap to attainable $V_{oc}$.

## Results

CdTe thin films were deposited either directly onto glass substrates or with varying thicknesses of CdSe underlayer present (intermixing of a CdSe/CdTe bilayer to form CdSe$_x$Te$_{1-x}$ is conventional in solar cell fabrication), both with and without post-deposition Cl passivation. Cl passivation was included as an additional variable due to its known influence on grain boundary passivation[26,27] and its ability to modify interdiffusion[28], while we avoided additional passivation agents to avoid overcomplicating the system. Simple glass substrates, as opposed to full device stacks, allowed us to directly examine the passivation properties of Se in CdSe$_x$Te$_{1-x}$ thin films in isolation. Cell structure samples (i.e. fabricated on typical electron/hole transport layers) exhibiting strong luminescence can be indicative of either better passivation or weaker charge extraction. Thus, producing samples on glass allows a cleaner analysis of passivation (see further discussion in Supplementary Information Note 1). This sample

structure is not without challenge due to the lack of surface roughness typically provided by underlying layers. Following chloride treatment, samples were found to have a degree of non-uniformity on the microscale. Specifically, the chloride treatment resulted in delamination of some sample regions from the substrate while others were left intact. This is due to the low roughness of the glass substrate, compared to typical underlying layers, limiting the adhesion of the CdTe film during recrystallization which occurs during Cl treatment. As an example of delamination, Fig. 1a shows a white light reflection image of a Cl-treated pure CdTe sample, i.e. with a 0 nm CdSe underlayer. Due to the spatially resolved nature of the measurements used, analysis was able to focus solely on areas where grains remained intact.

We carried out a number of morphological measurements to better understand the quality of the fabricated films, with results presented in Supplementary Information Note 2. Firstly, we carried out secondary ion mass spectrometry (SIMS) on samples with 50 nm and 200 nm CdSe underlayer to better understand the vertical distribution of Se throughout the films. In contrast to devices[29], we find that the ratio of Se to Te is approximately constant through the majority of the film, with a < 0.5 μm region close to the glass substrate containing a small increase in Se content, making these films close to ideal for spectroscopic analyses, as is presented in Fig. 1b as a function of SIMS etch number (see Supplementary Information Note 2 for further details). These measurements also confirmed that thicker CdSe underlayers resulted in greater Se content throughout the films. Secondly, we used interferometry measurements to record the thickness of our films, which we found to be between 2 μm and 2.5 μm in all cases. Finally, we used atomic force microscopy (AFM) to demonstrate that in non-delaminated regions, similar grain morphologies and surface roughness was observed for all chlorine-treated samples. We note for these measurements, and all others presented in the paper, multiple positions were recorded on measured samples with representative results being presented in the main text and Supplementary Information.

Microscale transmission and reflection measurements were carried out on all samples, which allowed us to calculate the fraction of light absorbed or scattered to high angles by the sample (equal to $1 - transmission - reflection$). Initially, we used a hyperspectral imaging system to measure across a wide sample area (see Supplementary Information Note 3). However, unexpectedly strong sub-bandgap inter-grain scattering was observed in all samples. To confirm whether these features were correct, we recorded the same signals along a lineslice (dashed line in Fig. 1a) in a second measurement system (see Methods). While this second approach limits the number of points measured, it has previously reproduced absorption coefficients from single crystals to an extremely high level of accuracy[30]. We present results from a Cl-treated 0 nm CdSe sample in Fig. 1c. We found extremely similar results in both cases, confirming strong sub-bandgap scattering (discussed further below). We present further results from the second measurement approach in the main text.

The deposited CdSe layers were expected to be completely consumed by interdiffusion with CdTe, resulting in CdSe$_x$Te$_{1-x}$ films with higher Se content for thicker CdSe layers. We carried out Tauc plot fitting (with sub-bandgap scattering subtracted) of the spatially averaged optical absorption data (see Supplementary Information Note 4) to determine the average bandgap of each sample[31]. The bandgap reduced with increased CdSe thickness, both with and without prior Cl treatment, as is presented in Fig. 1d. This demonstrates that samples behave as expected and we reproduce previous literature results well[5].

A spatially and spectrally resolved absorption map for a Cl-treated 0 nm CdSe sample is presented in Fig. 1c, alongside an image marking the sample region measured (for other samples, see Supplementary Information Note 4). Above the bandgap, strong absorption is observed at all wavelengths and positions, as expected for any good solar cell material. For grain centres absorption is never stronger than 0.75, which is reasonable noting the air-CdTe Fresnel power reflection

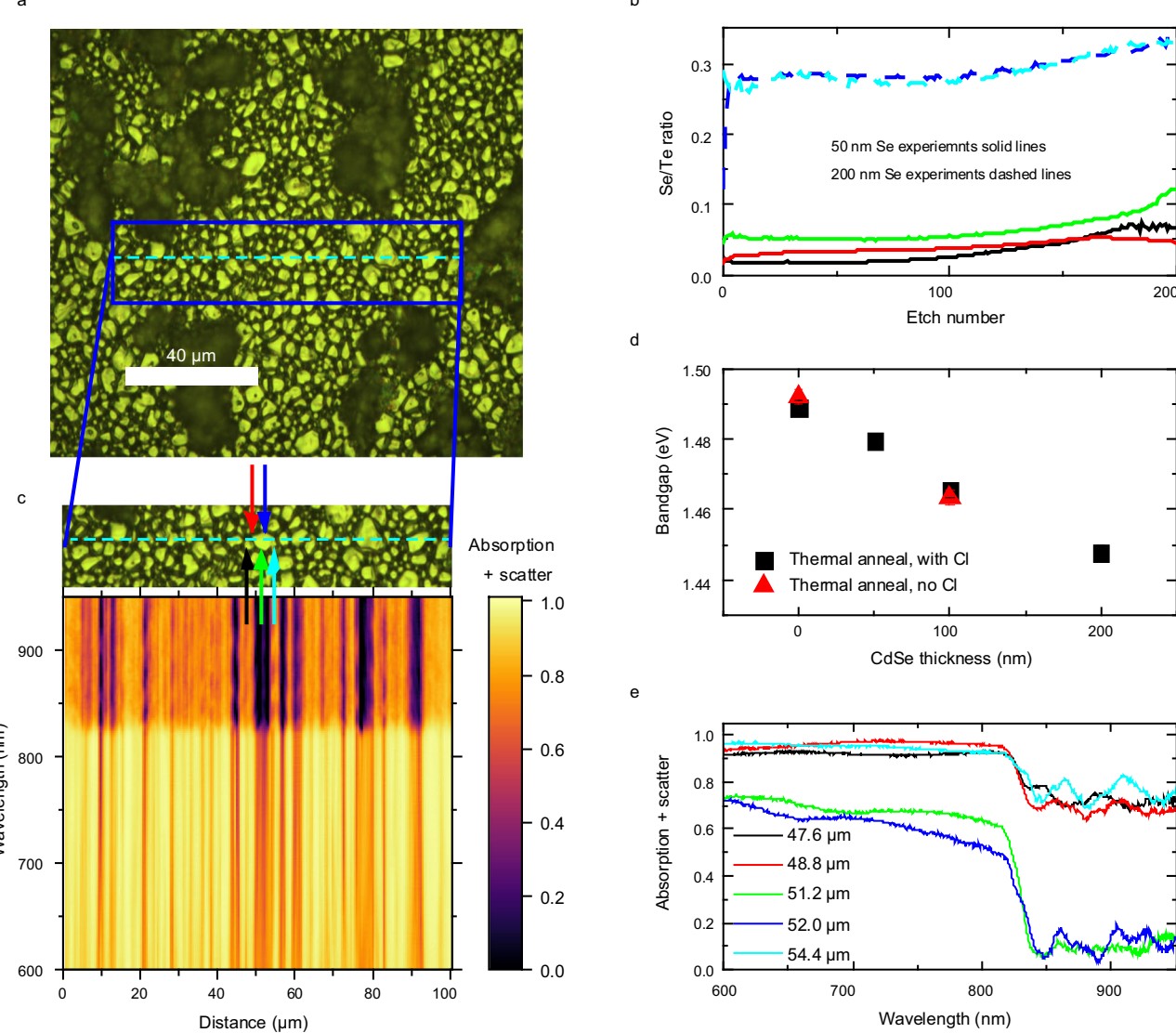

**Fig. 1 | Sample absorption and SIMS data. a** White light reflection image of a Cl-treated 0 nm CdSe sample. Extended dark areas correspond to region of delamination. Overlaid on this plot is a lineslice along which we measured sample absorption. **b** Ratio of Se to Te content as a function of etch number, as recorded via secondary ion mass spectrometry. Etch number 200 approximately corresponds to the glass substrate and each line to a different measurement (see legend). **c** Absorption + scatter results obtained for a 0 nm CdSe, Cl-treated sample. Above the main plot is the white light image of the sample with a blue dashed line highlighting the region measured. **d** Change in bandgap with CdSe thickness, via Tauc fitting of average absorption spectra from microscopic measurements, for samples both with and without a prior Cl-treatment. **e** Absorption + scatter for specific distances, as marked with colour arrows for specific positions in (**c**), as the measurement moves across a grain. Weaker below-bandgap scattering is seen when measuring on a grain (corresponding to 51.2 μm and 52.0 μm).

coefficient is -0.25. Below-bandgap (>820 nm) scattering reaches values greater than 0.5 between some grains but is close to 0 at grain centres. This is further demonstrated in Fig. 1e, where *absorption + scatter* is plotted for specific positions across a grain – high below-bandgap values are seen only between grains. Therefore, inter-grain regions scatter light into angles higher than 44° (based on the microscope objective NA of 0.7, see methods). We found that the extent of this inter-grain scattering was similar in all samples (Supplementary Information Note 4), implying scattering is not the main cause of changes to luminescence observed in subsequent parts of this work.

We attempted to use our absorption measurements to probe for bandgap changes across the sample – specifically whether we could correlate higher quantities of Se with lower bandgaps at grain boundaries. This was done by subtracting below-bandgap scattering from all samples and performing a Tauc fit at each position individually (see Supplementary Information Note 5). Larger bandgap variation was observed for samples containing more Se, and the data is suggestive of

lower bandgaps near gran boundaries. However, due to the strong variation in below bandgap scattering across the sample (Fig. 1c) it was difficult to draw a full conclusion, due to the error associated with the Tauc fitting (see Supplementary Information Note 5 for more detail).

To further explore the role of Se in our samples and explore passivation effects, we employed confocal mapping to record PL across the sample surface. We present average PL spectra for all Cl-treated samples of various Se content samples, as well as for 0 nm and 100 nm CdSe untreated samples, in Fig. 2a. We selected the incident laser intensity ($5.9 \times 10^6$ mW cm⁻²) to be high enough to allow for efficient mapping, but still within a trap-dominated regime (see Supplementary Information Note 6 for further discussion and evidence that the luminescence shape does not change with incident intensity). For all samples, a single main luminescence peak is seen. Interestingly, the PL peak wavelength of non-Cl treated samples shows no correlation with the quantity of CdSe present, despite Tauc fitting of the absorption measurements in Fig. 1d showing a reduced bandgap for the non-Cl treated 100 nm CdSe sample.

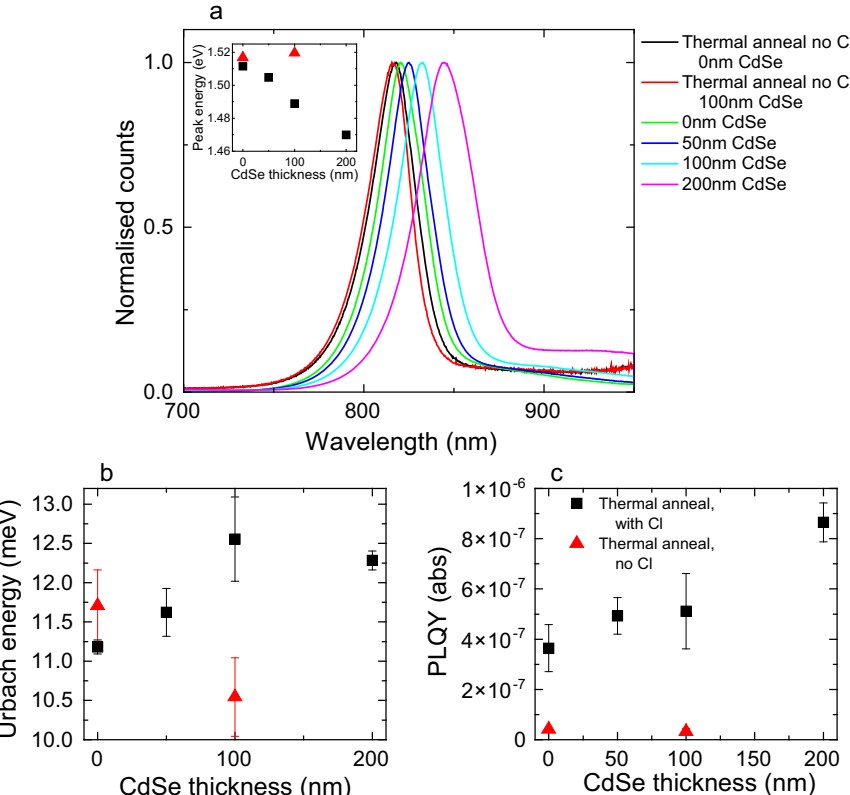

**Fig. 2 | Spatially averaged luminescence results. a** Spatially averaged photoluminescence, PL, observed from each map. The inset presents the peak PL position for each sample. **b** Spatially averaged Urbach energy for each sample, as a function of CdSe thickness. **c** Spatially averaged photoluminescence quantum yield (PLQY) as a function of sample thickness (for all luminescence wavelengths recorded). Legend in (**c**) applies to inset of (**a**–**c**). Error bars show standard deviation of recorded data.

This leads us to suggest that, without Cl-treatment, either the CdSe$_x$Te$_{1-x}$ phase absorbs light but is non-emissive, or charge diffusion between regions richer in CdTe (that absorb the majority of the light incident on the sample, as this side is excited) and phases with higher Se content is minimal. In either case, only PL from the pure CdTe grains is observed. For Cl-treated samples the PL peak does shift to lower energies with increasing Se content, and the peak corresponds well with bandgaps estimated from Tauc fitting (Fig. 2a, inset). In fact, for Cl-treated samples we were able to predict the sample luminescence well based on our measurements of sample absorption via the van Roosbroeck-Shockley relation[32], implying minimal Stokes shift prior to luminescence, as we discuss further in Supplementary Information Note 7. For higher quantities of Se-incorporation in Cl-treated samples, the PL not only red-shifts, but the width of the peak also increases (full-width-half-maximum of 41 nm for 200 nm CdSe thickness, compared to 34 nm for 0 nm CdSe thickness). This again suggests that there is greater bandgap variation in samples with higher Se content. Disorder within the main luminescence peak, as measured by fitting the PL with an Urbach tail (see Urbach fits in Supplementary Information Note 8)[33], is relatively constant for all samples, varying between 10 meV and 13 meV at most, as is presented in Fig. 2b. More surprisingly, as the quantity of Se is increased in Cl-treated samples, a longer, sub-bandgap wavelength (>900 nm) signal increases in relative intensity. This can be seen more clearly in Supplementary Information Note 8, where we present data on a logarithmic scale (to 950 nm) and uncalibrated data to 1050 nm (noting we were only confident in our radiometric calibration to 950 nm). This indicates that as Se content is increased, a defect state with strong luminescence increases in strength. As already noted, similar effects have been observed in bulk photoluminescence measurements by Kuciauskas et al. and Hu et al.[13,14], and cathodoluminescence measurements by Frouin et al.[12], but this is, to the best of our knowledge, the first photoluminescence evidence of a

clear correlation between a controlled Se content and the strength of the defect luminescence.

Figure 2c presents the spatially averaged photoluminescence quantum yield (PLQY) as a function of CdSe thickness. The PLQY measures of the ratio of radiative to total recombination rates – a perfect solar cell can only be achieved with a PLQY of unity. Here we observe PLQYs on the order of $10^{-7}$ – $10^{-6}$. Previous studies have reported PLQY values as high as $10^{-2}$ for surface-passivated CdSe$_x$Te$_{1-x}$ devices[34,35]. However, this drops to between $10^{-5}$ and $10^{-7}$ without As or Cu surface passivation[34], noting other surface passivations have only been observed to have weak effects on PLQY values[36]. Similarly, considering full devices, the best open circuit voltages observed in full devices is ~ 0.9 V while the detailed balance limit for CdTe predicts $V_{OC,ideal} = 1.14$ V[3,37]. Noting $V_{OC} = V_{OC,ideal} + \frac{k_B T}{q} ln(PLQY)$[38,39], this implies the best CdTe devices have PLQY ~ $10^{-3}$[36,40], but again these higher values are only observed with As or Cu surface passivation. Thus, the PLQY we observe is reasonable for samples without surface passivation that have been fabricated on glass, where surface recombination velocity would be expected to be higher than for an interface with optimised carrier extraction layers. Figure 2c confirms two additional points. Firstly, without Cl treatment the PLQY is approximately an order of magnitude lower, demonstrating this treatment is necessary for higher quality CdTe and that Se alone is not sufficient for passivation. Secondly, the PLQY also increases with increased Se content (we observe an approximate doubling of PLQY by 200 nm of CdSe), demonstrating that, while it does not preclude the need for a Cl treatment, CdSe plays a key role in film passivation.

Our PLQY values also allow us to consider where within our film our luminescence measurements probe. Specifically, at our excitation wavelength of 488 nm CdTe's absorption depth is 88 nm[41]. Charge diffusion lengths of up to ~ 30 μm have been observed in

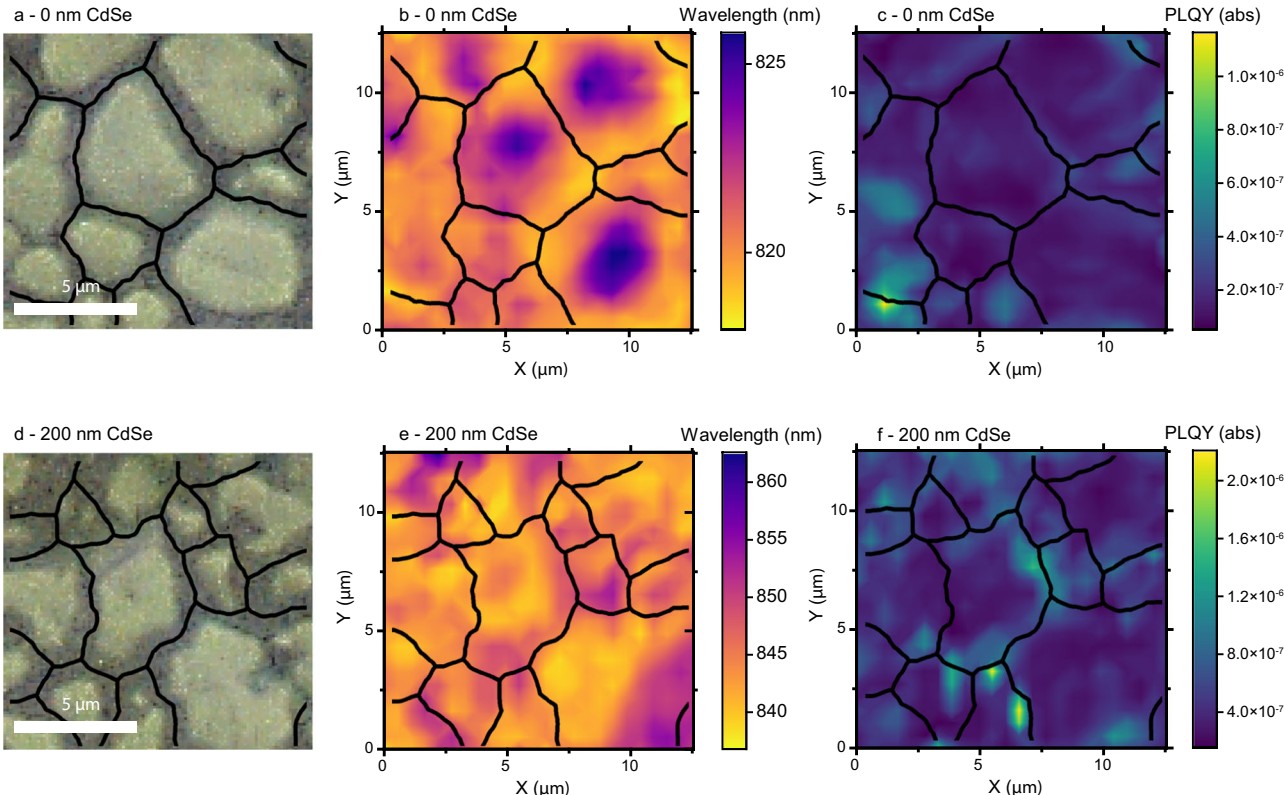

**Fig. 3 | Microscopic photoluminescence maps. a–c/d–f** White light reflection, peak wavelength and peak PLQY plots (±20 nm about the central peak) for 0 nm CdSe/200 nm CdSe samples, with both samples Cl treated. We note that the same colour scales are used for different ranges in (**b**)/(**e**) and (**c**)/(**f**) to allow for better visualisation of results. As a guide to the eye, black lines present approximate grain boundaries.

single crystals of CdTe[42] and as noted above the best PLQY results reported are on the order of $10^{-2}$ [35]. As our PLQY values of samples on glass are at least four orders of magnitude lower, and PLQY is (approximately) inversely proportional to charge lifetime, an upper bound on charge diffusion lengths in our films is 1 µm. Therefore, as the half of our films closer to air have a uniform Se to Te ratio (Supplementary Information Note 2), our photoluminescence signals originate from a region where the ratio of Se to Te can be considered to be constant. We plot this explicitly in Supplementary Information Fig. 1.

Having established the bulk CdSe$_x$Te$_{1-x}$ PL properties, we now consider microscale maps. Figure 3a–c/d–f shows a white light reflection image, peak wavelength map and local PLQY map for Cl-treated 0 nm/200 nm CdSe films. The peak wavelength is obtained by fitting the PL data from each pixel with a Gaussian, and details of PLQY measurement approach is provided in the methods section. Equivalent results for 50 nm and 100 nm CdSe films are provided in Supplementary Information Note 9 along with maps for non-Cl treated films. We note previous microscale studies of CdTe-based samples have not explored peak wavelength with position or directly quantified microscale PLQY[15–18]. Our spatially resolved analysis allows direct comparison of emission properties across the grain structure thereby distinguishing between grain boundaries and grain interiors. For Cl-treated 0 nm CdSe, longer wavelength PL emission peaks (> 822 nm) are observed predominantly from grain centres (Fig. 3a, b). As absorption measurements showed no significant bandgap variation across the sample (Supplementary Information Note 5), we interpret this to be a consequence of photon re-absorption i.e. shorter wavelength PL is re-absorbed before light can escape the grain, shifting the PL peak to longer wavelength from grain centres (as the PL must travel through more material before reaching the surroundings[43]). A direct consequence would be a higher PLQY from grain boundaries, which is observed in Fig. 3c, especially surrounding larger grains (for plots of overlaid grains and PLQY, see Supplementary Information Note 10). Notably, we also see some grain-to-grain variation in the PLQY – some grains (especially in the bottom left of Fig. 3c) are seen to have higher PLQYs. Grain-to-grain variation in PLQY is typical in many other semiconductor systems and can be related to numerous factors including strain and stoichiometry[44].

The situation is completely reversed for Cl-treated 200 nm CdSe films. Here we observe longer peak wavelengths predominantly at grain boundaries and shorter peak wavelengths in grain interiors (Fig. 3d/e, see Supplementary Information Note 10 for overlays). This is the opposite of what we expect from photon re-absorption, so it leads us to the conclusion (consistent with the greater variation of bandgap observed in absorption measurements) this must be a result of bandgap variation across the sample: grain edges must have lower energy bandgaps than grain centres, as has previously been observed in cross-sectional CL of CdSe$_x$Te$_{1-x}$ devices[10,45]. We again note that surface roughness does not vary significantly between samples (Supplementary Information Note 2), so this is not due to out-coupling alone. Additionally, we see significantly higher PLQYs from the grain boundaries compared to the grain centres, as is shown in Fig. 3f. The variation in PLQY between grain boundary and grain centre is approximately a factor of 2 larger than for 0 nm CdSe, and as we found the scattering at grain boundaries to be comparable for both samples (Supplementary Information Note 4), this reveals that the PLQY is stronger at grain boundaries due to better passivation. These results provide direct evidence that Se is incorporated at the grain boundaries and that it passivates these regions. We further explore our photoluminescence maps in Supplementary Information Note 11, where we extract the local temperature, radiative and implied open circuit voltages from our data. We find that all samples have uniform temperature distributions slightly above room temperature, which is

reasonable as we are using a focused laser beam to excite the sample. Our open circuit voltage maps further support our conclusions drawn here, again showing preferential passivation of grain boundaries for the 200 nm CdSe sample.

Cl-treated 50 nm CdSe and 100 nm CdSe plots show similar relationships between peak wavelength and PLQY to those presented in Fig. 3, as shown in Supplementary Information Note 9, with 50 nm CdSe behaving similarly to 0 nm CdSe and 100 nm CdSe more similarly to 200 nm CdSe. To further elucidate on this point (Supplementary Information Note 12), a negative correlation (Pearson correlation coefficient, $r = -0.13$) is observed between peak wavelength and PLQY for the 0 nm CdSe sample i.e. for longer wavelength regions (grain centres), the PLQY is lower (due to more photon re-absorption). However, as the quantity of CdSe is increased we observe a change to positive correlations ($r = 0.04, 0.18$ and $0.31$ for 50 nm, 100 nm and 200 nm CdSe respectively): longer wavelength regions have higher PLQYs. This again demonstrates that Se-rich regions (which have lower energy bandgaps) have better passivation. Due to much smaller grain sizes in non-Cl treated samples it was not possible to draw conclusions on the effect of grain structure on the PL, but we note we also see significant peak wavelength ($\pm 12$ nm) and PLQY variation (1 order of magnitude) across these samples.

Finally, we return to the below bandgap PL that increases with thicker CdSe (c.f. Figure 2a) to understand its origin at the microscale. We note that while our samples may not have optimal Se intermixing for device performance, the trend we see is consistent with increasing Se content across several regions measured, meaning the correlation between below-bandgap traps and increasing Se content is justifiable. For this defect to be appropriately passivated, its location within the sample must first be understood (going beyond the cathodoluminescence analyses by Frouin et al.[12]). Some below bandgap PL was observed across nearly all sample regions (as shown in Supplementary Information Note 13). To find where this emission was most pronounced relative to the main PL peak, we calculated PLQY maps of the long wavelength, below bandgap PL region (-900–950 nm) and the tail of the main PL peak (specifically, the range 10–20 nm beyond the main PL peak, focusing on these wavelengths to remove photon re-absorption effects that would complicate analysis). Normalising these maps by their average value and dividing the resulting long-wavelength data by the main PL peak tail data gives us a ratio that, if larger than one, indicates a region with particularly strong below-bandgap PL relative to its emission at the main PL peak wavelength. This ratio and associated white light reflection images for Cl-treated 0 nm CdSe/200 nm CdSe are shown in Fig. 4a, b/c, d. For 0 nm CdSe we observe a weak correlation between increased below-bandgap PL and grain boundaries (Fig. 4b), and the ratio stays between 0.7 and 1.3 for all positions, indicating only a modest difference between grain interiors and grain boundaries. In contrast, for 200 nm CdSe we see a strong increase in long wavelength PL (with the ratio reaching values larger than 2), mainly focused at grain centres. This leads us to the conclusion that while Se alloying passivates grain boundaries, its presence also correlates with an increase in an emissive defect state more concentrated within the grain centres, which would ultimately be detrimental to solar cell performance. Whilst the bandgap of the 200 nm CdSe sample indicates a lower Se content than typically found in record-efficiency devices[2], the trend towards increased sub-bandgap emission for higher Se content in this study suggests that this emissive state is likely to be prevalent in material with larger Se content. This is especially important in CdSe$_x$Te$_{1-x}$ solar cells where the ratio of Se to Te is graded across the cell. Hence these results imply that while Se incorporation helps to compensate for the deleterious impact of grain boundaries within the material, it may hinder further improvements by enhancing the formation of a luminescent trap state within the grains. Future work to explore this effect should explore longer wavelength photoluminescence, potentially using a Raman microscope with an InGaAs detectors (that can access

wavelengths longer than 1050 nm) coupled to short wavelength excitation (noting Raman microscopes with InGaAs detectors currently use long wavelength, >900 nm, excitation lasers).

This work showcases the possibilities of fully quantitative spatially resolved steady-state spectroscopy to understand CdTe and CdSe$_x$Te$_{1-x}$, providing new insight into the role of Se incorporation in CdSe$_x$Te$_{1-x}$. A tailored series of CdSe$_x$Te$_{1-x}$ films on glass were studied, where the Se content was varied via controlled thickness of a CdSe underlayer. Our microscale absorption and photoluminescence measurements give direct evidence that higher Se content passivates grain boundaries in CdSe$_x$Te$_{1-x}$, corroborating prior results from cross-sectional CL studies. Importantly, we reveal that Se alloying leads to an increase of luminescent sub-bandgap states, which are more strongly concentrated in grain centres. Our results indicate that the defect chemistry associated with Se is more complex than has been previously appreciated and likely has both beneficial and detrimental effects on the absorber material. Further work is now required to understand the precise nature of this defect state and whether its formation can be avoided by modification of the fabrication process.

## Methods
### Sample fabrication
CdSe$_x$Te$_{1-x}$ films with varying Se content were prepared on glass, without any conductive layers, in order to eliminate the effects of carrier extraction on the luminescence results. Soda-lime glass microscope slides were cut to 2.5 cm by 2.5 cm squares and cleaned by: scrubbing with a 1% Hellmanex™ solution by volume, a 15 min ultrasonic bath in de-ionised H$_2$O, then sequential rinses in de-ionised H$_2$O, acetone and propan-2-ol. This was followed by 10 min in an Ossila UV Ozone Cleaner to remove remaining organics from the surface. 50 nm, 100 nm and 200 nm CdSe layers were deposited on the substrates by radio-frequency sputtering with a power density of 1.32 W/cm$^2$, Ar pressure of 5 mTorr and a substrate temperature of 200 °C. These thicknesses were selected based on previous reports of CdSe layers thicker than 200 nm leading to unconsumed, photoinactive CdSe in devices[46,47]. CdTe was deposited by close-space sublimation (CSS), at a deposition pressure of 5 Torr in an N$_2$ atmosphere, with a source temperature of 600 °C and a substrate temperature of 550 °C. A 20 min post-growth anneal was performed in the CSS chamber immediately after deposition, at a substrate temperature of 550 °C and a pressure of 400 Torr. Film thicknesses were found to be between 2 μm and 2.5 μm from interferometry measurements. A Cl treatment was performed on some of the samples by placing them in a tube furnace adjacent to glass plates coated with 1 mol/dm$^3$ aqueous MgCl$_2$ solution, and annealing in air at 410 °C for 20 min. Excess chloride was rinsed off with de-ionised H$_2$O.

The CdSe and CdTe layers, whilst deposited separately, are expected to interdiffuse during the post-growth CSS anneal and Cl treatment steps, resulting in a CdSe$_x$Te$_{1-x}$ film[48]. It was found that substrate adhesion was particularly poor for the films deposited on glass. As such, a standard Cl treatment, involving spray-coating the MgCl$_2$ solution onto the samples before annealing, resulted in unacceptable levels of sample delamination from the substrate. Various Cl treatment conditions were tested, with the approach used in this work providing the optimal balance between Cl treatment passivation and reduced sample delamination. We note that our current fabrication approach was not able to access samples with higher CdSe thickness due to additional delamination problems.

### Absorption
Measurements (referred to as the 'second measurement system' in the text) were recorded on an NT&C system (a Nikon Eclipse Ti2 coupled to a Princeton Instruments Spectra Pro HRS-500 spectrometer and a PIXIS 256 camera). The light source for transmission measurements was a Halogen bulb and for reflection an Energetica LDLS™ laser driven white light source output through a fibre with 100 m core diameter.

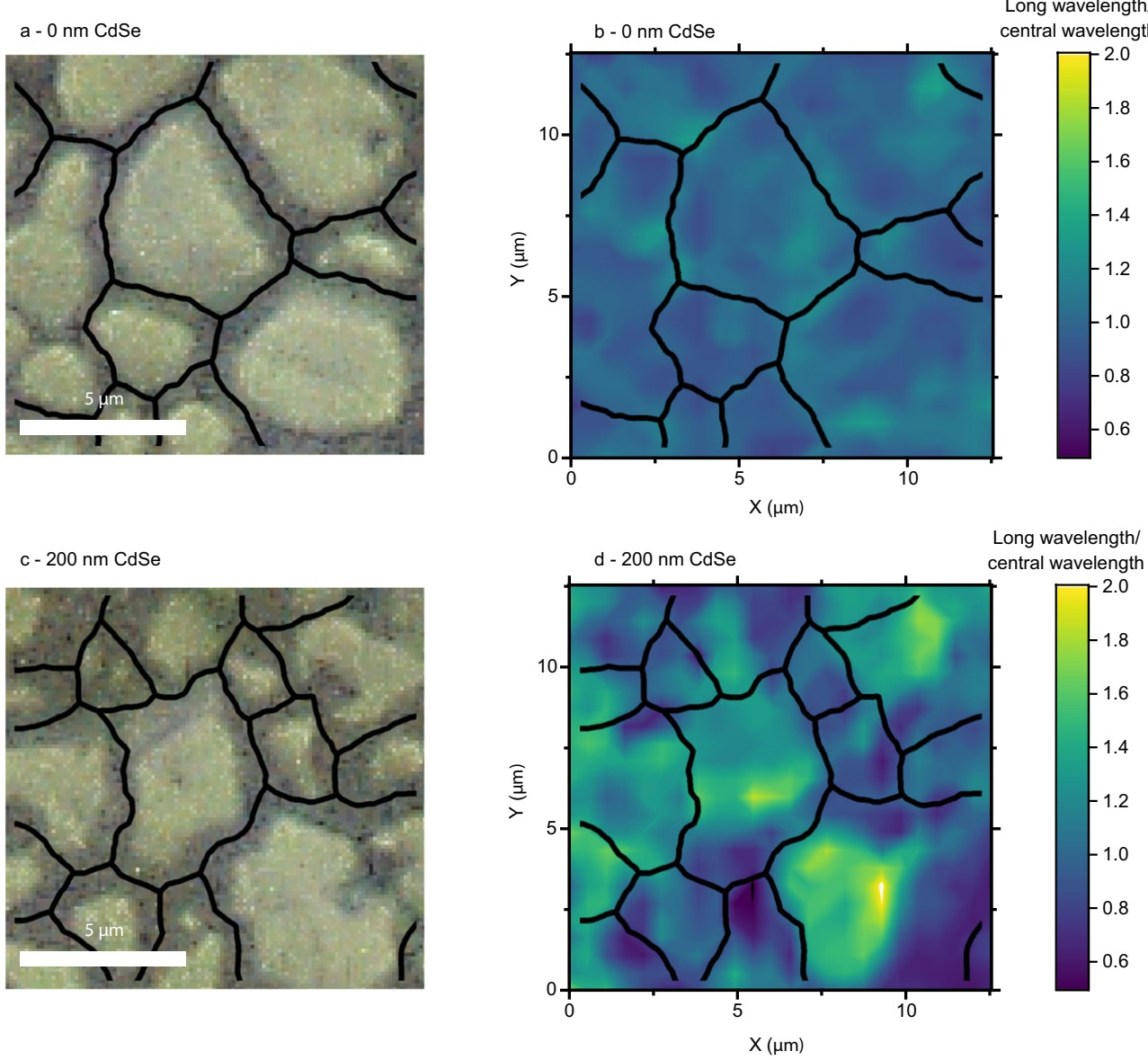

**Fig. 4 | Long-wavelength photoluminescence maps. a/c** White light reflection and (**b**)/(**d**) long wavelength PL intensity normalised to average value divided by the main wavelength PL intensity normalised to average value (see main text), for Cl-treated 0 nm/200 nm CdSe. As a guide to the eye, black lines present approximate grain boundaries.

For accurate transmission/reflection measurements, light should be incident perpendicular to the sample. To achieve this, for transmission measurements incident light was focused on the sample through a condenser lens with variable a-stop (almost fully closed), while for reflection incident light was focused in the centre of the objective back focal plane. The area illuminated was > 150 μm in all cases and transmission/reflection measurements were performed in a subregion within the illuminated spot (this avoids issues with inhomogeneous camera heating). Measurements were taken simultaneously over a lineslice by closing the slit at the entrance of the spectrometer to isolate the region of interest. The spectrometer camera, which was aligned at a conjugate focal plane, thus recorded the spectrum versus position in the region of interest without the need for a moving stage. In all measurements, transmitted or reflected light was collected by a 60x Nikon S Plan Fluor objective. All recorded signals were passed through a 450 nm long pass filter (Thorlabs FELH0450) to prevent second order effects on the spectrometer. For transmission a bare quartz substrate was used as a reference, while for reflection a mirror

of known spectral response (Thorlabs PF10-03-P10-10) was employed as the reference (the Thorlabs reported spectral response was used).

**Photoluminescence and photoluminescence quantum yield**
PL and PLQY were carried out on a Renishaw inVia Raman Microscope RE04 using a Coherent sapphire 488 SF NX excitation, incident on the air-$CdSe_xTe_{1-x}$ interface as the thickness of the glass substrate prevented good focus on the glass-$CdSe_xTe_{1-x}$ interface. All maps were recorded via snake scanning of the diffraction limited spot across the sample using an MS20 stage with minimum step size of 100 nm, dynamic trajectory control via a high-speed microprocessor, via serial communication, and real-time positional tracking using linear encoder feedback. We note the sample stage moved rather than the spot in all cases. PLQY was estimated using a similar approach to that outlined by Frohna et al.[43]. Specifically, a calibrated light source (Ocean Optics HL-3P-INT-CAL) was coupled to an integrating sphere with known spectral response (Thorlabs 2P3/M). The output port of this sphere was aligned with the objective lens (Lecia N PLAN EPI, 100x objective, NA = 0.85) of the

Raman microscope and the spectrum recorded (allowing for relative radiometric calibration of PL data). The notch filter that normally removes the laser signal from the recording path was then removed, and the signal again recorded. The ratio of these signals gives the spectral response with and without the notch filter present. Secondly, the sphere was removed and the objective lens was focused on a mirror (Thorlabs PF10-03-P10-10) with a known spectral response. The laser normally used to excite the sample was then shone on the mirror (at low power) and the signal recorded on the spectrometer. Finally, the mirror is removed and the incident power at the same position recorded with a Thorlabs S170C or S130C power metre. By accounting for the reflection strength of the mirror, the ratio of number of photons incident on the mirror and the number recorded by the spectrometer can be calculated i.e. an absolute calibration at one wavelength. Using the recorded lamp spectrum, it is then possible to work out an absolute calibration at any wavelength both with the notch filter present and removed. This calibration allows for absolute luminescence yield measurements. We note that the relative radiometric response of the sphere was measured by sending a collimated white light beam into a second integrating sphere and recording the signal at the output port on an Ocean Insight Spectrometer FLAME-S-XR1. The sphere of interest was then coupled to the second integrating sphere and the collimated beam sent into the sphere of interest. The ratio of these two signals gives the spectral response of the integrating sphere used in measurements. An incident laser intensity of $5.9 \times 10^6$ mWcm$^{-2}$ was used in all measurements unless otherwise stated (noting this was a beam focused in a small area that we assume diffraction limited). Spot size here is defined as the boundary through which the intensity falls to $\frac{1}{e^2}$ of the maximum intensity. Intensity of the laser was selected to allow for rapid sample measurement while preventing sample degradation (as observed by a gradual drop in the sample's photoluminescence with time, which was seen at higher laser intensities). When calculating PLQY we assume Lambertian emission into a full $\pi$ hemisphere forwards and backwards, that is we multiply the number of recorded photons by $\frac{2}{NA^2}$, where $NA$ is the numerical aperture of the objective, equal to 0.85 in these measurements. Furthermore, we were unable to measure absorption from the same region as PL, so when calculating PLQY we used the spatially averaged absorption value of 0.9 (see Supplementary Fig. 4). We also note that we initially attempted PL measurements using wide-field illumination and hyperspectral mapping. In this hyperspectral system we found that signals from laminated regions were too weak to be recorded. This initiated our approach of using a Raman Microscope with a focused laser beam.

**Secondary ion mass spectrometry**
SIMS was carried out on an IMS 1280 HR from Cameca. A Cs+ primary beam, with +10 kV on the source and −10kV on the sample (20 kV of impact energy) was employed in all experiments, with a Max Area of 80 μm and electronic gating of 30%. We explored different experimental parameters, but for those presented in the main text, for 50 nm and 200 nm CdSe samples, we used a field aperture of 4000/3000 μm, 150 μm raster length, a mass resolving power of 3000 and a primary beam current of 2.2-3 (sample dependent)/2 nA. While our SIMS measurements record regions with both laminated and delaminated material, we note delamination occurs between the thin film and the glass rather than within the sample, meaning the Se/Te distribution should be the same in both regions. Additionally, multiple measurements were taken at different points across samples to verify the Se distribution was consistent and that measurements were not unduly influenced by any delamination.

**White light interferometry**
Interferometry was used to record sample thickness following SIMS. Specifically, a Bruker Contour GT-K was employed, using a white light source in VSI/VXI mode, with a 10x objective.

**Atomic force microscopy**
AFM was used to measure sample roughness. A Bruker FastScan AFM was employed for all measurements in ScanAsyst mode. ScanAsyst-Fluid+ tips were used in all measurements. 1st-order plane fits were applied to all results. We note that in two AFM scans (out of more than 16 scans in total), we observed a large number of very small (<1 μm size) grains nestled between larger grains. We removed these scans prior to the roughness analysis presented in the Supplementary Information.

## Data availability
The data underlying this manuscript and source data files are available at https://doi.org/10.5281/zenodo.13869384.

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

## Acknowledgements

A.R.B. acknowledges the support of an EPSRC Impact Accelerator Grant, and an SNSF Swiss Postdoctoral Fellowship TMPFP2_217040. G.T. and A.R.B. acknowledge the support of an SNSF Eccellenza Grant PCEGP2-194181. A.R.B. thanks Franky Esteban Bedoya Lora for the use of an Ocean Optics spectrometer. J.F.L. and J.D.M. acknowledge EPSRC for funding via EP/N014057/1, EP/W03445X/1 and the EPSRC Centre for Doctoral Training in New and Sustainable Photovoltaics EP/L01551X/1. K.F. acknowledges the support of an Engineering and Physical Sciences Research Council (EPSRC) Doctoral Prize Postdoctoral Fellowship and a Winton Sustainability Fellowship. S.D.S. acknowledges support from the Royal Society and Tata Group (UF150033). The authors acknowledge the EPSRC (EP/R023980/1) for funding. The authors thank Florent Plane, Anne-Sophie Bouvier and Johanna Marin Carbonne for their assistance with SIMS measurements.

## Author contributions

A.R.B. carried out confocal luminescence mapping, absorption maps displayed in the main text, and AFM measurements. ARB coordinated SIMS measurements and assisted in hyperspectral mapping measurements. A.R.B. carried out all luminescence data analysis and co-wrote the text. J.F.L. conceived the project, fabricated all samples and co-wrote the text. K.F. carried out all hyperspectral mapping measurements with A.R.B. and assisted with data interpretation. S.D.S. supported data interpretation, G.T. and J.D.M. supervised the project and assisted with writing the text.

## Competing interests

The authors declare no competing interests.
