## [Peer Review File · Nature Communications]

Spatially resolved photoluminescence analysis of the role of Se
in $\text{CdSe}_x\text{Te}_{1-x}$ thin filmsREVIEWER COMMENTS

Reviewer #1 (Remarks to the Author):

Bowman et al use microscale PL and absorption measurements to map the photoluminescence quantum yield (PLQY) of selenium-alloyed CdTe films. Higher PLQY at grain boundaries for films containing selenium suggests that it passivates grain boundaries. Sub-bandgap emission from films containing selenium indicates that the element is associated with an emissive trap state.

The work is a significant advance in PL measurements of CdTe. Absolute PL measurements like performed in this paper have been used to great effect in loss analysis of perovskite solar cells (including in papers by some of the authors) and the community will welcome its application to CdTe. As a methods paper it will have significant impact. Areas that can be improved are listed below.

Suggested changes:

Change 1

Provide further explanation of the possible effects of charge extraction at contacts on PL measurements. At present it is mentioned on three separate occasions but not explained in detail.

For instance the authors state:

‘Notably, prior microscale PL studies have measured samples in full or partial device stacks. This presents a fundamental issue: it is impossible to disentangle spectroscopically whether changes in charge lifetime or luminescence strength originate from charge extraction or sample passivation processes’

And

‘Cell structure samples (i.e. fabricated on typical electron/hole transport layers) exhibiting strong luminescence can be indicative of either better passivation or weaker charge extraction. Thus, producing samples on glass allows a cleaner analysis of passivation.’

To what extent can the excited charges be ‘extracted’ at the contacts if the cell is at open circuit? Since carriers can’t leave the cell are they stored in the contacts throughout the measurement? The text would benefit from a more detailed explanation of the proposed mechanism at play here along with references of previous work describing the mechanism. The authors go to great lengths to make the films on glass rather than on partial cells, so this decision needs to be fully justified (issues associated with glass deposition that the authors encounter include absorber delamination, having to use a lighter than normal chlorine treatment, and having an absorber grown

on atypical surface i.e glass rather than contact layers). PLQY measurements are frequently done on absorber layers with contacts present to measure the effects of the contacts on recombination and hence internal voltage. If charge extraction from the absorber layer to the contacts occurs at open circuit, will it not affect the measurements described in the literature as well?

Change 2

The authors control Se content of the films by changing the thickness of the CdSe layer in as-deposited CdSe/CdTe bilayers. The bilayers are then annealed. During the anneal selenium from the CdSe layer diffuses to some extent into the CdTe. Some of the films then undergo a further anneal in the presence of chlorine (which can also affect Se intermixing/distributions). The distribution of Se in the films following these anneals is important for interpretation of the PL and absorption data. The paper would therefore benefit from measurement of cross sectional Se distributions in the films. In addition, as it stands there are conflicting statements about the selenium distributions that need to be clarified. Some sentences suggest that the selenium is homogeneously distributed throughout the films following annealing, whereas others say it is unevenly distributed.

For instance, the authors state that:

‘The deposited CdSe layers were expected to completely interdiffuse with CdTe, resulting in CdSe_xTe_{1-x} films, with higher Se content for thicker CdSe layers’

And:

‘Our results show that Se is distributed throughout the films’

However, later on the authors state:

‘...or charge diffusion between regions richer in CdTe (that absorb the majority of the light incident on the sample, as this side is excited) and phases with higher Se content is minimal. In either case, only PL from the pure CdTe grains is observed.’

The best way to remove this uncertainty is to map the Se distributions in cross-section. If there are variations in the vertical distributions of Se in the films, as suggested by the third statement, then this would be problematic. For instance if there is Se left at the bottom of the films (i.e. there is a u-shaped profile that is often seen with bilayer interdiffusion in polycrystalline films) then this makes interpretation of the backside PL and the through-thickness absorption measurements presented in the paper very difficult. This possibility needs to be ruled out.

Compositional mapping could also shed light on why the PL emission energy does not match the E_g estimated from the absorption measurements, as discussed by the authors on page 9.

Compositional Se mapping can be done using SEM-based or TEM-based EDX spectroscopy.

Change 3

Surprisingly, there is no attempt to determine the Se alloying fraction of the annealed films.

The authors should measure the selenium alloying % in the 'no chlorine' and chlorine treated films (EDX mapping would achieve this).

The Bg of the '0 nm CdSe' film is only 0.04 eV higher than the '200 nm CdSe' film. This suggests that there is a relatively small amount of Se present in the films (~5 at.% Se even in their most Se-rich film). This low concentration likely explains why the sub-bandgap emission peak is relatively small.

The optimum Se alloying percentage in CdTe absorbers is 10-20 at.%. This is what is used in most research cells as efficiencies are lower with lower Se content. If measurements confirm that the Se % in their films is indeed significantly lower than that typically used in CdSeTe solar cells then authors should comment on this difference and explain how the results for low Se % films might relate to the higher Se % films used in solar cells.

As done above the Se content of the films can be estimated indirectly using the bandgap measurements (either using the absorption edge or PL emission peak). However, there are issues using this method if the Se is not homogeneously distributed throughout the absorber layer, as illustrated by the discrepancy between the Bg from absorption measurements and the peak emission energy described on page 9.

I therefore emphasise that the authors should measure the Se content and distributions of the films directly using a technique such as EDX spectroscopy.

Change 4

On Fig 1b change to legend label to: 'no Cl, purely thermal anneal' or something similar, making clear that the sample is thermally annealed and not left as-deposited.

Reviewer #2 (Remarks to the Author):

Bowman et al report spectrally resolved PL analysis of CdSeTe. I appreciate authors effort in this relatively small research area. But there are three major errors in the study that should preclude publication (anywhere):

1) Authors need to report films' chemical composition - and spatial uniformity of chemical composition - using chemical analysis (SIMS, AES, or other). Absorption and PL data indicate

bandgaps of 1.49-1.44 eV. (Fig. 1, Fig. S6, Fig. S9). Using published bowing relationships in CdSeTe (e.g., J. Yang and S.-H. Wei, *Chin. Phys. B* 28, 086106 (2019)), Se composition in samples under study is very low. In contrast, ~40 atomic % Se are used in high efficiency solar cells, and bandgap in high efficiency CdSeTe solar cells is 1.40 eV (e.g., R. Mallick et al, *IEEE J. Photovoltaics* 13, 510 (2023) – this paper reports characterization of world-record Cd(Se)Se solar cell, including it is PLQY, implied and radiative voltage, cross-sectional CL, and probably needs to be cited). Thus, authors are analyzing absorbers not relevant for high efficiency solar cells. Also, with delamination shown by the authors (Fig. 1) it is likely that Se composition is very non-uniform.

2) Unfortunately, absorbers are of low electronic quality – PLQY is $\sim E^{-7}$ (Fig. 2), while it is $\sim E^{-3}$ in high quality absorbers and solar cells (Mallick et al. above, Kuciauskas 2020 – 2023, Onno 2022 references in the manuscript). Electronic quality needs to be more fully characterized – need to report radiative voltage, implied voltage, and PL measurement temperature (e.g., from fit to the high energy side of the spectrum). Some of the authors are experts in this analysis – Frohna 2022 is an excellent paper. While this data is not given, from spectra it appears that temperature is very high – e.g. manual shift was used in Figure S9, and PL in Figure 1a extends much too far to the high energy (short wavelength) side. Authors need to make better absorbers, and study them with more care.

3) Spectral range of PL measurements need to extend to lower energies than used in this study. E.g., as reported by Neupane 2023, Kuciauskas 2023, CdSeTe can have emission down to 0.8 eV (1600 nm), and such defect states can severely impact electronic properties. Data for increased Se compositions (which are still too low!) in Fig. 1 (200 nm CdSe), Fig. S8, Fig. S10 strongly suggest that measurements with InGaAs detectors (or similar) are also needed. Until that measurement is used, the topic in the manuscript title is not addressed.

4) There are additional more specific questions about the data and analysis, but they are not needed at this stage.

Reviewer #3 (Remarks to the Author):

1. The intrinsic physical mechanism of the impact of the introduction of Se on the photoelectric properties of CdSeTe thin films is very important for the development of high-efficiency CdTe thin film solar cells. The author used micro-area PL and other technologies to study the impact of Se introduction in CdSeTe on its properties. No similar research has been reported in the literature so far. This paper is of great reference for researchers working on CdTe thin film solar cells and is worthy of publication.

2. However, in order to ensure that the short-circuit current density of the device is sufficiently high, when preparing CdTe solar cells with an absorption layer including CdSeTe, specific processes are usually adopted to control the Se composition to obtain a CdSeTe layer with a bandgap of ~ 1.39 eV. In fact, the QE curve of the CdTe solar cell with the highest photoelectric conversion efficiency reported in the literature shows that its long wavelength cutoff edge extends to 1.39 eV [Green MA, Dunlop ED, Yoshita M, et al. *Solar cell efficiency tables (Version 63)*. *Prog Photovolt Res Appl.* 2023;1-11. doi:10.1002/pip.3750.]. However, the band gap of the sample studied in this article is

only ~1.44 eV. In order to better understand the intrinsic mechanism by which the introduction of Se affects the properties of CdSeTe films, it is recommended that the authors provide characterization results of samples with bandgaps ranging from 1.45 eV to 1.39 eV.

3. In addition, in order to help readers better understand and reproduce the experimental results of the manuscript, the author should supplement more details on the preparation process and characterization techniques of certain samples, such as:

(1) The relevant details of the "Photoluminescence and Photoluminescence Quantum Yield (PLQY)" test, such as the size or spatial resolution of the excited light spot, as well as the control mechanism of the movable sample stage used and the spatial resolution of the movable sample stage.

(2) Details related to the "Microscale transmission and reflection measurements" test, such as the size of the spot after the incident light is focused on the surface of the sample, and whether a sample moving table was also used during the testing process? The manuscript mentions "due to spatial and noise resolution limits in our measurements, we were unable to draw definitive conclusions from absorption measurements all". Please provide a detailed explanation.

4. The sample preparation technique used in this manuscript may result in a gradually decreasing distribution of Se content from the substrate to the interface between the film and air. When conducting PL testing, the measured fluorescence intensity is related to the absorption length of the excitation light wavelength used (usually the reciprocal of the absorption coefficient), which means that the fluorescence intensity is a contribution of a certain depth range of CdSeTe samples with non-uniform Se distribution. The manuscript should analyze whether the vertical distribution of Se affects its conclusion. If possible, the manuscript should provide characterization results of the longitudinal Se distribution of the sample.

5. Usually, it is challenging to obtain the thin and dense CdTe films with using CSS technology. The manuscript should provide more details concerning CSS process and the thickness of the prepared samples.

Reviewer #4 (Remarks to the Author):

The stack is glass/CdSe/CdTe. There is no passivation layer in the front or the back leading to a high degree of recombination in the front and back surfaces. This makes it difficult to draw conclusions about passivation in the bulk/grain boundaries.

The CdCl₂ treatment led to delamination and the treatment was not optimum leaving defects in the film and difficult to draw conclusions about the nature of PL. Also, the CdCl₂ treatment did not change the bandgap indicating very low intermixing.

Due to the low CdCl₂ treatment the redistribution of Se is likely to be limited and high non-uniform Se concentration and making it difficult to study the effects of Se.

As many reported earlier, Se diffuses through grain boundaries into the bulk. Figure 3 e) indicates poor intermixing of CdSe/CdTe with low Se concentration in the bulk in comparison to grain

boundaries. Resulting in better PLQY in the grain boundaries than in the bulk.

Reviewer #5 (Remarks to the Author):

The stack is glass/CdSe/CdTe. There is no passivation layer in the front or the back leading to a high degree of recombination in the front and back surfaces. This makes it difficult to draw conclusions about passivation in the bulk/grain boundaries.

The CdCl₂ treatment led to delamination and the treatment was not optimum leaving defects in the film and difficult to draw conclusions about the nature of PL. Also, the CdCl₂ treatment did not change the bandgap indicating very low intermixing.

Due to the low CdCl₂ treatment the redistribution of Se is likely to be limited and high non-uniform Se concentration and making it difficult to study the effects of Se.

As many reported earlier, Se diffuses through grain boundaries into the bulk. Figure 3 e) indicates poor intermixing of CdSe/CdTe with low Se concentration in the bulk in comparison to grain boundaries. Resulting in better PLQY in the grain boundaries than in the bulk.

Reviewer #1 (Remarks to the Author):

Bowman et al use microscale PL and absorption measurements to map the photoluminescence quantum yield (PLQY) of selenium-alloyed CdTe films. Higher PLQY at grain boundaries for films containing selenium suggests that it passivates grain boundaries. Sub-bandgap emission from films containing selenium indicates that that the element is associated with an emissive trap state.

The work is a significant advance in PL measurements of CdTe. Absolute PL measurements like performed in this paper have been used to great effect in loss analysis of perovskite solar cells (including in papers by some of the authors) and the community will welcome its application to CdTe. As a methods paper it will have significant impact. Areas that can be improved are listed below.

We thank the reviewer for their comments and are delighted to learn that they consider it a positive contribution to the CdTe field.

Suggested changes:

Change 1

Provide further explanation of the possible effects of charge extraction at contacts on PL measurements. At present it is mentioned on three separate occasions but not explained in detail.

For instance the authors state:

‘Notably, prior microscale PL studies have measured samples in full or partial device stacks. This presents a fundamental issue: it is impossible to disentangle spectroscopically whether changes in charge lifetime or luminescence strength originate from charge extraction or sample passivation processes’

And

‘Cell structure samples (i.e. fabricated on typical electron/hole transport layers) exhibiting strong luminescence can be indicative of either better passivation or weaker charge extraction. Thus, producing samples on glass allows a cleaner analysis of passivation.’

To what extent can the excited charges be ‘extracted’ at the contacts if the cell is at open circuit? Since carriers can’t leave the cell are they stored in the contacts throughout the measurement? The text would benefit from a more detailed explanation of the proposed mechanism at play here along with references of previous work describing the mechanism. The authors go to great lengths to make the films on glass rather than on partial cells, so this decision needs to be fully justified (issues associated with glass deposition that the authors encounter include absorber delamination, having to use a lighter than normal chlorine treatment, and having an absorber grown on atypical surface i.e glass rather than contact layers). PLQY measurements are frequently done on absorber layers with contacts present to measure the effects of the contacts on recombination and hence internal voltage. If charge extraction from the absorber layer to the contacts occurs at open circuit, will it not affect the measurements described in the literature as well?

We thank the reviewer for raising this important point. We have added an additional note in the supporting information specifically to discuss this point (which is appropriately linked to the main text):

“Supporting Information Note 1 – luminescence with charge extraction layers present

We study samples on glass to remove charge extraction effects from our spectroscopic results. Here we briefly discuss additional effects that can be observed with charge extraction layers or at interfaces

between two electronically active materials, noting that this is an area of ongoing spectroscopic research. Specifically, we give two examples from complementary fields.

Our first example is time resolved photoluminescence (TRPL) from halide perovskites. Several studies have focused on changes to TRPL signals when a halide perovskite is on glass compared to on a charge extraction layer, including Stolterfoht et al [1]. Here it was observed that a halide perovskite had significantly shorter TRPL lifetime when placed on a charge extraction layer (see figure 3 in this paper), noting that shorter TRPL lifetimes are equivalent to weaker luminescence signals in steady state luminescence. The authors attribute this reduced lifetime to a mixture of charge extraction and interfacial recombination. Importantly, while TRPL lifetimes are significantly longer following interfacial passivation (see figure 5c and 5f in this paper), there is still a rapid initial drop and charge lifetime is still lower than on glass. This shows that a reduction in TRPL lifetime (i.e. luminescence) can be due to charge extraction or interfacial charge traps, and it is difficult to deconvolute these competing effects. This has been further explored by a number of others in this field [2,3], generally demonstrating that charge transport layers can have a multitude of effects on the strength of luminescence signals. Notably, significant changes are seen even with a single transport layer deposited on the active layer, which is a form of open circuit (likely due to interfacial recombination and some charges being extracted to the transport layer and then subsequently undergoing non-radiative recombination across the interface).

Our second example is from organic semiconductors. Here it is common to study heterojunctions between two electronically active materials. Below-bandgap photoluminescence peaks have been observed due to charge transfer states between two organic semiconductors, for example as shown by Ng et al [4]. This demonstrates that when multiple electronically active materials are present their interaction can introduce interfacial luminescence states. By studying bare CdTe samples on glass we are able to rule out that the below-bandgap luminescence states seen in our system originate from a charge transfer state or similar.

Both these examples demonstrate the importance of studying samples on glass (or other inert substrate) when focusing on the effect of a single material within a solar cell device stack.”

[1] M. Stolterfoht et al., Visualization and Suppression of Interfacial Recombination for High-Efficiency Large-Area Pin Perovskite Solar Cells, *Nat Energy* 3, 847 (2018).

[2] L. Krückemeier, Z. Liu, B. Krogmeier, U. Rau, and T. Kirchartz, Consistent Interpretation of Electrical and Optical Transients in Halide Perovskite Layers and Solar Cells, *Adv. Energy Mater.* 11, 2102290 (2021).

[3] L. Krückemeier, B. Krogmeier, Z. Liu, U. Rau, and T. Kirchartz, Understanding Transient Photoluminescence in Halide Perovskite Layer Stacks and Solar Cells, *Advanced Energy Materials* 11, 2003489 (2021).

[4] A. M. C. Ng, A. B. Djurišić, W.-K. Chan, and J.-M. Nunzi, Near Infrared Emission in Rubrene:Fullerene Heterojunction Devices, *Chemical Physics Letters* 474, 141 (2009).

Change 2

The authors control Se content of the films by changing the thickness of the CdSe layer in as-deposited CdSe/CdTe bilayers. The bilayers are then annealed. During the anneal selenium from the CdSe layer diffuses to some extent into the CdTe. Some of the films then undergo a further anneal in the presence of chlorine (which can also affect Se intermixing/distributions). The distribution of Se in the films following these anneals is important for interpretation of the PL and absorption data. The paper would therefore benefit from measurement of cross sectional Se distributions in the films. In addition, as it stands there are conflicting statements about the selenium distributions that need to be clarified. Some sentences suggest that the selenium is homogeneously distributed throughout the films following annealing, whereas others say it is unevenly distributed.

For instance, the authors state that:

‘The deposited CdSe layers were expected to completely interdiffuse with CdTe, resulting in Cd_{1-x}Se_xTe films, with higher Se content for thicker CdSe layers’

And:

‘Our results show that Se is distributed throughout the films’

However, later on the authors state:

‘...or charge diffusion between regions richer in CdTe (that absorb the majority of the light incident on the sample, as this side is excited) and phases with higher Se content is minimal. In either case, only PL from the pure CdTe grains is observed.’

We have removed contrasting statements from the text, following our additional measurements (see discussions below). However, we note this final sentence noted by the reviewer is specifically for non-Cl treated samples.

The best way to remove this uncertainty is to map the Se distributions in cross-section. If there are variations in the vertical distributions of Se in the films, as suggested by the third statement, then this would be problematic. For instance if there is Se left at the bottom of the films (i.e. there is a u-shaped profile that is often seen with bilayer interdiffusion in polycrystalline films) then this makes interpretation of the backside PL and the through-thickness absorption measurements presented in the paper very difficult. This possibility needs to be ruled out.

Compositional mapping could also shed light on why the PL emission energy does not match the E_g estimated from the absorption measurements, as discussed by the authors on page 9.

Compositional Se mapping can be done using SEM-based or TEM-based EDX spectroscopy.

Following this comment we were inspired to carry out secondary ion mass spectrometry (SIMS) to measure the distribution of Se, Te and the ratio of the two from the front to the back of the film. We note that other approaches suggested above were challenging due to sample charging effects. Our SIMS measurements have allowed us to clarify statements regarding the distribution of Se and Te throughout the film in the main text, add the following paragraph to the main text regarding SIMS:

“We carried out a number of morphological measurements to better understand the quality of the fabricated films, with results presented in Supporting Information note 2. Firstly, we carried out secondary ion mass spectrometry (SIMS) on samples with 50 nm and 200 nm CdSe underlayer to better understand the vertical distribution of Se throughout the films. In contrast to devices [Artegiani2021], we find that the ratio of Se to Te is approximately constant through the majority of the film, with a $< 0.5 \mu\text{m}$ region close to the glass substrate containing a small increase in Se content, making these films close to ideal for spectroscopic analyses, as is presented in Figure 1b as a function of SIMS etch number (see Supporting Information note 2 for further details). These measurements also confirmed that thicker CdSe underlayers resulted in greater Se content throughout the films. Secondly, we used interferometry measurements to record the thickness of our films, which we found to be between $2 \mu\text{m}$ and $2.5 \mu\text{m}$ in all cases. Finally, we used atomic force microscopy to demonstrate that in non-delaminated regions, similar grain morphologies and surface roughnesses were observed for all chlorine-treated samples. We note for these measurements, and all others presented in the paper, multiple positions were recorded on measured samples with representative results being presented in the main text and supporting information.

Figure 1. a) white light reflection image of a Cl-treated 0 nm CdSe sample. Extended dark areas correspond to region of delamination. Overlaid on this plot is a lineslice along which we measured sample absorption. b) Ratio of Se to Te content as a function of etch number, as recorded via secondary ion mass spectrometry. Etch number 200 approximately corresponds to the glass substrate at the back of the film and each line to a different measurement (see legend). c) absorption + scatter results obtained for a 0 nm CdSe, Cl-treated sample. Above the main plot is the white light image of the sample with a blue dashed line highlighting the region measured. d) change in bandgap with CdSe thickness, via Tauc fitting of average absorption spectra from microscopic measurements, for samples both with and without a prior Cl-treatment. e) absorption + scatter for specific distances, as marked with colour arrows for specific positions in c), as the measurement moves across a grain. Weaker below-bandgap scattering is seen when measuring on a grain (corresponding to 51.2 μm and 52.0 μm).

[Artegianni2021] E. Artegianni, P. Punathil, V. Kumar, M. Bertocello, M. Meneghini, A. Gasparotto, and A. Romeo, Effects of CdTe Selenization on the Electrical Properties of the Absorber for the Fabrication of $\text{CdSe}_x\text{Te}_{1-x}/\text{CdTe}$ Based Solar Cells, *Solar Energy* 227, 8 (2021).

And the following text to the supporting information:

“SIMS was used to record the vertical distribution of Se and Te throughout films. Specifically, we studied samples fabricated with 50 nm CdSe and 200 nm CdSe underlayers (noting the time intensive nature of this measurement prevented us from also exploring 100 nm CdSe samples). We recorded the intensity of Se and Te present in the film, then used a Cs^+ ion beam to remove a thin layer of the film repeatedly (see methods), building a vertical profile. This process was repeated until we reached the substrate, following previous analyses of CdSeTe samples [5]. We present our results in Figure S1, for three regions measured in the 50 nm CdSe sample and two regions measured in the 200 nm CdSe sample, as a function of the etch number in the measurement. The absolute intensities of the Se and Te

signals were found to vary between samples, as is presented in Figure S1a and S1b, as is expected for samples with different surfaces exposed and small variations in measurement parameters between each experiment. However, the key quantity is the ratio of Se to Te, which we present in Figure S1c. This reveals three important points: i) the Se/Te ratio is relatively consistent across different sample regions; ii) samples fabricated with 200 nm CdSe underlayers have significantly higher Se relative to Te; and iii) within each sample the ratio of Se to Te is relatively uniform throughout the film, with significant changes occurring only close to 200 etches i.e. extremely close to the glass substrate (see Figure S1a/b). This shows that the vast majority of our films have relatively uniform proportions of Se and Te and that ratio can be controlled via the thickness of the initial CdSe layer.

Figure S1. SIMS intensity for a) Se and b) Te as a function of Etch number (with 0 etches being the air-exposed surface). c) ratio of Se to Te intensities. Legend in c) applies to all plots.”

[5] E. Artegiani, P. Punathil, V. Kumar, M. Bertinello, M. Meneghini, A. Gasparotto, and A. Romeo, Effects of CdTe Selenization on the Electrical Properties of the Absorber for the Fabrication of CdSe_xTe_{1-x}/CdTe Based Solar Cells, *Solar Energy* 227, 8 (2021).”

Regarding where our PL signal originates from, we have added the following to the main text:

“Our PLQY values also allow us to consider where within our film our luminescence measurements probe. Specifically, at our excitation wavelength of 488 nm CdTe’s absorption depth is 88 nm [Treharne2011]. Charge diffusion lengths of up to $\sim 30 \mu\text{m}$ have been observed in single crystals of CdTe [Ščajev2022], and as noted above the best PLQY results reported are on the order of 10^{-2} [Kuciauskas2020]. As our PLQY values of samples on glass are at least four orders of magnitude lower, and PLQY is (approximately) inversely proportional to charge lifetime, an upper bound on charge diffusion lengths in our films is $1 \mu\text{m}$. Therefore, as the front half of our films have a uniform Se to Te ratio (Supporting Information Note 2), our photoluminescence signals originate from a region where the ratio of Se to Te is constant.”

[Kuciauskas2020] D. Kuciauskas, J. Moseley, P. Ščajev, and D. Albin, Radiative Efficiency and Charge-Carrier Lifetimes and Diffusion Length in Polycrystalline CdSeTe Heterostructures, *Phys. Status Solidi RRL*, 14(3), 1900606 (2020)

[Ščajev2022] Ščajev, P., Mekys, A., Subačius, L. et al. Impact of dopant-induced band tails on optical spectra, charge carrier transport, and dynamics in single-crystal CdTe. *Scientific Reports*, 12, 12851 (2022).

[Treharne2011] R. E. Treharne, A. Seymour-Pierce, K. Durose, K. Hutchings, S. Roncallo and D. Lane. Optical Design and Fabrication of Fully Sputtered CdTe/CdS Solar Cells. *Journal of Physics: Conference Series*. 286 012038 (2011)

Finally, we have addressed the mismatch between absorption and PL, with the supporting information note reading:

“Supporting Information Note 7 – prediction of photoluminescence from absorption measurements

We applied the van Roosbroeck-Shockley relation [7] to predict the spectral shape of the photoluminescence from our absorption measurements. Specifically, this relation states that $PL(E) \propto a(E)E^2 e^{-\frac{E}{k_B T}}$, where $a(E)$ is the measured absorption as a function of energy E and $k_B T$ the thermal energy, with the temperatures extracted in Supporting Information Note 11 used. We used the scattering subtracted, spatially averaged $a(E)$ (i.e. that presented in Figure S5) and found we had to shift predicted values up by 7 meV (i.e. predicted PL peaks were at lower energies than measured PL peaks) to obtain good agreement with spatially averaged measured photoluminescence. We attribute this shift to our PL measurements probing the front half of the CdTe layer (see main text for discussion) while absorption measurements probe the entire CdTe layer. As our SIMS results showed (supporting information note 2), near the back of the film is a small region with increased Se content, and thus marginally lower bandgap observed in absorption. Finally, we note that there is significant error in our below bandgap $a(E)$ measurements, so we only predict the energies close to the PL peak on the lower energy side. A comparison between experiment and theory is presented in Figure S10, with the good agreement implying that there is minimal Stokes shift prior to photoluminescence.

Figure S10. Comparison between measured (solid lines) and absorption predicted (dashed lines) photoluminescence for Cl-treated samples. Predicted PL was shifted by 7 meV to have good agreement (see text above).”

[7] W. van Roosbroeck and W. Shockley, Photon-Radiative Recombination of Electrons and Holes in Germanium, *Physical Review* 94, 1558 (1954).

Change 3

Surprisingly, there is no attempt to determine the Se alloying fraction of the annealed films. The authors should measure the selenium alloying % in the ‘no chlorine’ and chlorine treated films (EDX mapping would achieve this).

The Bg of the ‘0 nm CdSe’ film is only 0.04 eV higher than the ‘200 nm CdSe’ film. This suggests that there is a relatively small amount of Se present in the films (~5 at.% Se even in their most Se-rich film). This low concentration likely explains why the sub-bandgap emission peak is relatively small.

The optimum Se alloying percentage in CdTe absorbers is 10-20 at.%. This is what is used in most research cells as efficiencies are lower with lower Se content. If measurements confirm that the Se % in their films is indeed significantly lower than that typically used in CdSeTe solar cells then authors should comment on this difference and explain how the results for low Se % films might relate to the higher Se % films used in solar cells.

As done above the Se content of the films can be estimated indirectly using the bandgap measurements (either using the absorption edge or PL emission peak). However, there are issues using this method if the Se is not homogeneously distributed throughout the absorber layer, as illustrated by the discrepancy between the Bg from absorption measurements and the peak emission energy described on page 9.

I therefore emphasise that the authors should measure the Se content and distributions of the films directly using a technique such as EDX spectroscopy.

As noted above we have used secondary ion mass spectrometry (SIMS) to measure the distribution of Se content throughout the films. While this does not provide an absolute ratio of the Se to Te (we note EDX will can cause significant local charging issues as samples are not on a conductive substrate), it has allowed us to understand the distribution of Se throughout the films in more detail, as already discussed. Importantly, these results show a meaningful difference in the Se content in the films between the 50 nm CdSe sample and the 200 nm CdSe one. The SIMS profiles show constant compositions at the depths probed by PL measurements, and as such the position of the emission peak is a valid estimate of the Se content. Published bandgap bowing relationships for $\text{CdSe}_x\text{Te}_{1-x}$ indicate the bandgap reduction of 0.05 eV observed in this work corresponds to a Se content of 10 at% [J. Yang and S.-H. Wei, Chin. Phys. B 28, 086106 (2019)]. The following text has been added to the manuscript to comment on how our results could relate to material with higher Se content:

“Whilst the bandgap of the 200 nm CdSe sample indicates a lower Se content than typically found in record-efficiency devices [Green63], the trend towards increased sub-bandgap emission for higher Se content in this study suggests that this emissive state is likely to be prevalent in material with larger Se content.”

[Green63] M. A. Green, E. D. Dunlop, M. Yoshita, N. Kopidakis, K. Bothe, G. Siefer, and X. Hao, Solar cell efficiency tables (version 62), Progress in Photovoltaics: Research and Applications, 31(7), (2023)

Additionally, we have added the following to the text regarding the choice of these CdSe thicknesses when carrying out sample fabrication:

“These thicknesses were selected based on previous reports of CdSe layers thicker than 200 nm leading to unconsumed, photoinactive CdSe in devices [Poplawsky2016,Baines2018].”

[Poplawsky2016] J. D. Poplawsky, W. Guo, N. Paudel, A. Ng, K. More, D. Leonard and Y. Yan, Structural and Compositional Dependence of the $\text{CdTe}_x\text{Te}_{1-x}$ Alloy Layer Photoactivity in CdTe-Based Solar Cells, Nature Communications, 7(1), 12537 (2016)

[Baines2018] T. Baines, Z. Guillaume, L. Bowen, T. P. Shalvey, S. Mariotti, K. Durose and J. D. Major, Incorporation of CdSe Layers into CdTe Thin Film Solar Cells, Solar Energy Materials and Solar Cells, 180, 196-204 (2018)

Change 4

On Fig 1b change to legend label to: 'no Cl, purely thermal anneal' or something similar, making clear that the sample is thermally annealed and not left as-deposited.

We agree and have updated relevant figures as suggested, which we present below:

“

Figure 1. a) white light reflection image of a Cl-treated 0 nm CdSe sample. Extended dark areas correspond to region of delamination. Overlaid on this plot is a lineslice along which we measured sample absorption. b) Ratio of Se to Te content as a function of etch number, as recorded via secondary ion mass spectrometry. Etch number 200 approximately corresponds to the glass substrate at the back of the film and each line to a different measurement (see legend). c) absorption + scatter results obtained for a 0 nm CdSe, Cl-treated sample. Above the main plot is the white light image of the sample with a blue dashed line highlighting the region measured. d) change in bandgap with CdSe thickness, via Tauc fitting of average absorption spectra from microscopic measurements, for samples both with and without a prior Cl-treatment. e) absorption + scatter for specific distances, as marked with colour arrows for specific positions in c), as the measurement moves across a grain. Weaker below-bandgap scattering is seen when measuring on a grain (corresponding to 51.2 μm and 52.0 μm).

Figure 2. a) spatially averaged photoluminescence, PL, observed from each map. The inset presents the peak PL position for each sample. b) spatially averaged Urbach energy for each sample, as a function of CdSe thickness. c) spatially averaged photoluminescence quantum yield (PLQY) as a function of sample thickness (for all luminescence wavelengths recorded). Legend in c) applies to inset of a), b) and c).”

Reviewer #2 (Remarks to the Author):

Bowman et al report spectrally resolved PL analysis of CdSeTe. I appreciate authors effort in this relatively small research area.

We thank the reviewer for their comments, which we address below. While we agree that compared to some fields CdTe is a small research area, we note that it is one of only two commercially competitive solar cell technologies, so we believe that research in this area carries significant impact.

But there are three major errors in the study that should preclude publication (anywhere):

1) Authors need to report films' chemical composition - and spatial uniformity of chemical composition - using chemical analysis (SIMS, AES, or other). Absorption and PL data indicate bandgaps of 1.49-1.44 eV. (Fig. 1, Fig. S6, Fig. S9). Using published bowing relationships in CdSeTe (e.g., J. Yang and S.-H. Wei, Chin. Phys. B 28, 086106 (2019)), Se composition in samples under study is very low. In contrast, ~40 atomic % Se are used in high efficiency solar cells, and bandgap in high efficiency CdSeTe solar cells is 1.40 eV (e.g., R. Mallick et al, IEEE J. Photovoltaics 13, 510 (2023) – this paper reports characterization of world-record Cd(Se)Se solar cell, including it is PLQY, implied and radiative voltage, cross-sectional CL, and probably needs to be cited). Thus, authors are analyzing absorbers not relevant for high efficiency solar cells. Also, with delamination shown by the authors (Fig. 1) it is likely that Se composition is very non-uniform.

We thank the reviewer for raising this point. Firstly, the change in bandgap in our samples (from Tauc fitting) is 0.05 eV. Using the bowing relationship highlighted by the reviewer, this corresponds to our samples increasing to approximately 10 atomic % Se, which we believe is a significant shift in the Se content enabling meaningful comparisons between samples and establish trends in the material's properties with increasing Se content. This work focusses on preparing samples that enable the cleanest possible analysis of the effect of Se on the material, rather than mimicking record-efficiency device structures, which we note typically involve graded compositions that would complicate interpretation of the absorption and PL results. Since the impact of fabrication on glass on the CdSe/CdTe interdiffusion was not known, we used CdSe thicknesses similar to previous studies on full devices [Baines2018] in an effort to avoid unconsumed CdSe near the glass interface. This has resulted in lower peak Se content due to the more uniform intermixing in our samples, but we note that thicker CdSe layers would have exacerbated issues with film delamination. We have added the following to the text regarding this point:

“These thicknesses were selected based on previous reports of CdSe layers thicker than 200 nm leading to unconsumed, photoinactive CdSe in devices [Poplawsky2016,Baines2018].”

[Poplawsky2016] J. D. Poplawsky, W. Guo, N. Paudel, A. Ng, K. More, D. Leonard and Y. Yan, Structural and Compositional Dependence of the CdTe_xTe_{1-x} Alloy Layer Photoactivity in CdTe-Based Solar Cells, Nature Communications, 7(1), 12537 (2016)

[Baines2018] T. Baines, Z. Guillaume, L. Bowen, T. P. Shalvey, S. Mariotti, K. Durose and J. D. Major, Incorporation of CdSe Layers into CdTe Thin Film Solar Cells, Solar Energy Materials and Solar Cells, 180, 196-204 (2018)

We note R. Mallick's paper was already cited in our work in the section discussing PLQY, which we highlight here:

“Here we observe PLQYs on the order of 10⁻⁷ - 10⁻⁶. Previous studies have reported PLQY values as high as 10⁻² for surface-passivated CdSe_xTe_{1-x} devices[Onno2022,Kuciauskas2020]. However, this drops to between 10⁻⁵ and 10⁻⁷ without As or Cu surface passivation[Onno2022], noting other surface passivations have only been observed to have weak effects on PLQY values [Mallick2023]. Similarly, considering full devices, the best open circuit voltages observed in full devices is ~ 0.9 V while the detailed balance limit for CdTe predicts $V_{OC,ideal} = 1.14$ V[Geisthardt2015,Ruhle2016]. Noting $V_{OC} =$

$V_{OC,ideal} + \frac{k_B T}{q} \ln(PLQY)$ [Ross1967, Miller2012], this implies the best CdTe devices have $PLQY \sim 10^{-3}$ [Scarpulla2023, Mallick2023], but again these higher values are only observed with As or Cu surface passivation.”

[Geisthardt2015] R. M. Geisthardt, M. Topič and J. R. Sites. Status and Potential of CdTe Solar-Cell Efficiency. *IEEE Journal of Photovoltaics*, 5, 4, (2015).

[Kuciauskas2020] D. Kuciauskas, J. Moseley, P. Ščajev, and D. Albin, Radiative Efficiency and Charge-Carrier Lifetimes and Diffusion Length in Polycrystalline CdSeTe Heterostructures, *Phys. Status Solidi RRL*, 14(3), 1900606 (2020)

[Mallick2023] R. Mallick, X. Li, X. Shan et al. Arsenic-Doped CdSeTe Solar Cells Achieve World Record 22.3% Efficiency. *IEEE Journal of Photovoltaics*, 13, 4 (2023)

[Miller2012] O. D. Miller, E. Yablonovitch, and S. R. Kurtz, Strong Internal and External Luminescence as Solar Cells Approach the Shockley–Queisser Limit, *IEEE J. Photovoltaics* 2, 303 (2012)

[Onno2022] A. Onno, C. Reich, S. Li, A. Danielson, W. Weigand, A. Bothwell, S. Grover, J. Bailey, G. Xiong, D. Kuciauskas, W. Sampath, and Z. C. Holman, Understanding What Limits the Voltage of Polycrystalline CdSeTe Solar Cells, *Nat. Energy*, 7, 400-408 (2022)

[Ross1967] R. T. Ross, Some Thermodynamics of Photochemical Systems, *The Journal of Chemical Physics* 46, 4590 (1967)

[Ruhle2016] S. Rühle, Tabulated values of the Shockley-Queisser limit for single junction solar cells, *Solar Energy*, 130, 139-147 (2016)

[Scarpulla2023] M. A. Scarpulla, B. McCandless, A. B. Phillips *et al.*, CdTe-based thin film photovoltaics: Recent advances, current challenges and future prospects, *Solar Energy Materials and Solar Cells*, 255, 112289 (2023)

Additionally, we note that high efficiency CdTeSe solar cells have been reported with peak Se contents <20 at% [Li2022], with the average Se content of the film lower still due to the graded composition.

[Li2022] D.-B. Li, S. S. Bista, R. A. Awni, S. Neupane, A. Abudulimu, X. Wang, K. K. Subedi, M. K. Jamarkattel, A. B. Phillips, M. J. Heben, J. D. Poplawsky, D. A. Cullen, R. J. Ellingson and Y. Yan, 20%-Efficient Polycrystalline Cd(Se,Te) Thin-Film Solar Cells with Compositional Gradient Near the Front Junction, *Nature Communications*, 13, 7849 (2022)

Finally, we have now carried out secondary ion mass spectrometry which has demonstrated that Se is in fact uniformly distributed through the sample. We have added the following to the text:

“We carried out a number of morphological measurements to better understand the quality of the fabricated films, with results presented in Supporting Information note 2. Firstly, we carried out secondary ion mass spectrometry (SIMS) on samples with 50 nm and 200 nm CdSe underlayer to better understand the vertical distribution of Se throughout the films. In contrast to devices [Artegiani2021], we find that the ratio of Se to Te is approximately constant through the majority of the film, with a < 0.5 μm region close to the glass substrate containing a small increase in Se content, making these films close to ideal for spectroscopic analyses, as is presented in Figure 1b as a function of SIMS etch number (see Supporting Information note 2 for further details). These measurements also confirmed that thicker CdSe underlayers resulted in greater Se content throughout the films. Secondly, we used interferometry measurements to record the thickness of our films, which we found to be between 2 μm and 2.5 μm in all cases. Finally, we used atomic force microscopy to demonstrate that in non-delaminated regions, similar grain morphologies and surface roughnesses were observed for all chlorine-treated samples. We note for these measurements, and all others presented in the paper, multiple positions were recorded on measured samples with representative results being presented in the main text and supporting information.

Figure 1. a) white light reflection image of a Cl-treated 0 nm CdSe sample. Extended dark areas correspond to region of delamination. Overlaid on this plot is a lineslice along which we measured sample absorption. b) Ratio of Se to Te content as a function of etch number, as recorded via secondary ion mass spectrometry. Etch number 200 approximately corresponds to the glass substrate at the back of the film and each line to a different measurement (see legend). c) absorption + scatter results obtained for a 0 nm CdSe, Cl-treated sample. Above the main plot is the white light image of the sample with a blue dashed line highlighting the region measured. d) change in bandgap with CdSe thickness, via Tauc fitting of average absorption spectra from microscopic measurements, for samples both with and without a prior Cl-treatment. e) absorption + scatter for specific distances, as marked with colour arrows for specific positions in c), as the measurement moves across a grain. Weaker below-bandgap scattering is seen when measuring on a grain (corresponding to 51.2 μm and 52.0 μm).”

[Artegianni2021] E. Artegianni, P. Punathil, V. Kumar, M. Bertocello, M. Meneghini, A. Gasparotto, and A. Romeo, Effects of CdTe Selenization on the Electrical Properties of the Absorber for the Fabrication of $\text{CdSe}_x\text{Te}_{1-x}/\text{CdTe}$ Based Solar Cells, *Solar Energy* 227, 8 (2021).

And the following text to the supporting information:

“SIMS was used to record the vertical distribution of Se and Te throughout films. Specifically, we studied samples fabricated with 50 nm CdSe and 200 nm CdSe underlayers (noting the time intensive nature of this measurement prevented us from also exploring 100 nm CdSe samples). We recorded the intensity of Se and Te present in the film, then used a Cs^+ ion beam to remove a thin layer of the film repeatedly (see methods), building a vertical profile. This process was repeated until we reached the substrate, following previous analyses of CdSeTe samples [5]. We present our results in Figure S1, for three regions measured in the 50 nm CdSe sample and two regions measured in the 200 nm CdSe sample, as a function of the etch number in the measurement. The absolute intensities of the Se and Te

signals were found to vary between samples, as is presented in Figure S1a and S1b, as is expected for samples with different surfaces exposed and small variations in measurement parameters between each experiment. However, the key quantity is the ratio of Se to Te, which we present in Figure S1c. This reveals three important points: i) the Se/Te ratio is relatively consistent across different sample regions; ii) samples fabricated with 200 nm CdSe underlayers have significantly higher Se relative to Te; and iii) within each sample the ratio of Se to Te is relatively uniform throughout the film, with significant changes occurring only close to 200 etches i.e. extremely close to the glass substrate (see Figure S1a/b). This shows that the vast majority of our films have relatively uniform proportions of Se and Te and that ratio can be controlled via the thickness of the initial CdSe layer.

Figure S1. SIMS intensity for a) Se and b) Te as a function of Etch number (with 0 etches being the air-exposed surface). c) ratio of Se to Te intensities. Legend in c) applies to all plots.”

[5] E. Artegiani, P. Punathil, V. Kumar, M. Bertocello, M. Meneghini, A. Gasparotto, and A. Romeo, Effects of CdTe Selenization on the Electrical Properties of the Absorber for the Fabrication of CdSexTe1-x/CdTe Based Solar Cells, *Solar Energy* 227, 8 (2021).

Finally, we have added a discussion of exactly the region of the samples we are probing with PLQY:

“Our PLQY values also allow us to consider where within our film our luminescence measurements probe. Specifically, at our excitation wavelength of 488 nm CdTe’s absorption depth is 88 nm [Treharne2011]. Charge diffusion lengths of up to ~ 30 μm have been observed in single crystals of CdTe [Ščajev2022], and as noted above the best PLQY results reported are on the order of 10-2 [Kuciauskas2020]. As our PLQY values of samples on glass are at least four orders of magnitude lower, and PLQY is (approximately) inversely proportional to charge lifetime, an upper bound on charge diffusion lengths in our films is 1 μm . Therefore, as the front half of our films have a uniform Se to Te ratio (Supporting Information Note 2), our photoluminescence signals originate from a region where the ratio of Se to Te is constant.”

[Kuciauskas2020] D. Kuciauskas, J. Moseley, P. Ščajev, and D. Albin, Radiative Efficiency and Charge-Carrier Lifetimes and Diffusion Length in Polycrystalline CdSeTe Heterostructures, *Phys. Status Solidi RRL*, 14(3), 1900606 (2020)

[Ščajev2022] Ščajev, P., Mekys, A., Subačius, L. et al. Impact of dopant-induced band tails on optical spectra, charge carrier transport, and dynamics in single-crystal CdTe. *Scientific Reports*, 12, 12851 (2022).

[Treharne2011] R. E. Treharne, A. Seymour-Pierce, K. Durose, K. Hutchings, S. Roncallo and D. Lane. Optical Design and Fabrication of Fully Sputtered CdTe/CdS Solar Cells. *Journal of Physics: Conference Series*. 286 012038 (2011)

2) Unfortunately, absorbers are of low electronic quality – PLQY is ~E-7 (Fig. 2), while it is ~E-3 in high quality absorbers and solar cells (Mallick et al. above, Kuciauskas 2020 – 2023, Onno 2022 references in the manuscript). Electronic quality needs to be more fully characterized – need to report radiative voltage, implied voltage, and PL measurement temperature (e.g., from fit to the high energy side of the spectrum). Some of the authors are experts in this analysis – Frohna 2022 is an excellent paper. While this data is not given, from spectra it appears that temperature is very high – e.g. manual

shift was used in Figure S9, and PL in Figure 1a extends much too far to the high energy (short wavelength) side. Authors need to make better absorbers, and study them with more care.

Our study specifically focuses on the role of Se in thin films without any competing effects from electrical contacts or additional passivation layers, as we now extensively discuss in a new Supplementary Information note 1:

“Supporting Information Note 1 – luminescence with charge extraction layers present

We study samples on glass to remove charge extraction effects from our spectroscopic results. Here we briefly discuss additional effects that can be observed with charge extraction layers or at interfaces between two electronically active materials, noting that this is an area of ongoing spectroscopic research. Specifically, we give two examples from complementary fields.

Our first example is time resolved photoluminescence (TRPL) from halide perovskites. Several studies have focused on changes to TRPL signals when a halide perovskite is on glass compared to on a charge extraction layer, including Stolterfoht et al [1]. Here it was observed that a halide perovskite had significantly shorter TRPL lifetime when placed on a charge extraction layer (see figure 3 in this paper), noting that shorter TRPL lifetimes are equivalent to weaker luminescence signals in steady state decays. The authors attribute this reduced lifetime to a mixture of charge extraction and interfacial recombination. Importantly, while TRPL lifetimes are significantly longer following interfacial passivation (see figure 5c and 5f in this paper), there is still a rapid initial drop and charge lifetime is still lower than on glass. This shows that a reduction in TRPL lifetime (i.e. luminescence) can be due to charge extraction or interfacial charge traps, and it is difficult to deconvolute these competing effects. This has been further explored by a number of others in this field [2,3], generally demonstrating that charge transport layers can have a multitude of effects on the strength of luminescence signals. Notably, significant changes are seen even with a single transport layer deposited on the active layer, which is a form of open circuit (likely due to interfacial recombination and some charges being extracted to the transport layer and then subsequently undergoing non-radiative recombination across the interface).

Our second example is from organic semiconductors. Here it is common to study heterojunctions between two electronically active materials. Below-bandgap photoluminescence peaks have been observed due to charge transfer states between two organic semiconductors, for example as shown by Ng et al [4]. This demonstrates that when multiple electronically active materials are present their interaction can introduce interfacial luminescence states. By studying bare CdTe samples on glass we are able to rule out that the below-bandgap luminescence states seen in our system originate from a charge transfer state or similar.

Both these examples demonstrate the importance of studying samples on glass (or other inert substrate) when focusing on the effect of a single material within a solar cell device stack.”

[1] M. Stolterfoht et al., Visualization and Suppression of Interfacial Recombination for High-Efficiency Large-Area Pin Perovskite Solar Cells, *Nat Energy* 3, 847 (2018).

[2] L. Krückemeier, Z. Liu, B. Krogmeier, U. Rau, and T. Kirchartz, Consistent Interpretation of Electrical and Optical Transients in Halide Perovskite Layers and Solar Cells, *Adv. Energy Mater.* 11, 2102290 (2021).

[3] L. Krückemeier, B. Krogmeier, Z. Liu, U. Rau, and T. Kirchartz, Understanding Transient Photoluminescence in Halide Perovskite Layer Stacks and Solar Cells, *Advanced Energy Materials* 11, 2003489 (2021).

[4] A. M. C. Ng, A. B. Djurišić, W.-K. Chan, and J.-M. Nunzi, Near Infrared Emission in Rubrene:Fullerene Heterojunction Devices, *Chemical Physics Letters* 474, 141 (2009).

As this is one of the first attempts at fabrication of CdTe on glass and we do not use passivation layers, we do not find it surprising that the PLQY is lower than world record efficiencies. However, we note that in the works highlighted above by the reviewer, higher PLQY values are only observed with

additional Cu or As passivation, while PLQY values of bare CdSe_xTe_{1-x} are extremely comparable with the values we observe (see for example [Kuciauskas2023] or [Onno2022], where PLQY values on the order of 10⁻⁶ to 10⁻⁷ are reported for undoped films. We have clarified this point in the main text as follows:

“Here we observe PLQYs on the order of 10⁻⁷ - 10⁻⁶. Previous studies have reported PLQY values as high as 10⁻² for surface-passivated CdSe_xTe_{1-x} devices[Onno2022,Kuciauskas2020]. However, this drops to between 10⁻⁵ and 10⁻⁷ without As or Cu surface passivation[Onno2022], noting other surface passivations have only been observed to have weak effects on PLQY values [Mallick2023]. Similarly, considering full devices, the best open circuit voltages observed in full devices is ~ 0.9 V while the detailed balance limit for CdTe predicts $V_{OC,ideal} = 1.14$ V[Geisthardt2015,Ruhle2016]. Noting $V_{OC} = V_{OC,ideal} + \frac{k_B T}{q} \ln(PLQY)$ [Ross1967,Miller2012], this implies the best CdTe devices have PLQY ~ 10⁻³ [Scarpulla2023,Mallick2023], but again these higher values are only observed with As or Cu surface passivation.”

[Geisthardt2015] R. M. Geisthardt, M. Topič and J. R. Sites. Status and Potential of CdTe Solar-Cell Efficiency. IEEE Journal of Photovoltaics, 5, 4, (2015)

[Kuciauskas2020] D. Kuciauskas, J. Moseley, P. Ščajev, and D. Albin, Radiative Efficiency and Charge-Carrier Lifetimes and Diffusion Length in Polycrystalline CdSeTe Heterostructures, Phys. Status Solidi RRL, 14(3), 1900606 (2020)

[Kuciauskas2023] D. Kuciauskas, M. Nardone, A. Bothwell, D. Albin, C. Reich, C. Lee, E. Colegrove. Why Increased CdSeTe Charge Carrier Lifetimes and Radiative Efficiencies did not Result in Voltage Boost for CdTe Solar Cells. Adv. Energy Mater. 13, 2301784 (2023)

[Mallick2023] R. Mallick, X. Li, X. Shan et al. Arsenic-Doped CdSeTe Solar Cells Achieve World Record 22.3% Efficiency. IEEE Journal of Photovoltaics, 13, 4 (2023)

[Miller2012] O. D. Miller, E. Yablonovitch, and S. R. Kurtz, Strong Internal and External Luminescence as Solar Cells Approach the Shockley–Queisser Limit, IEEE J. Photovoltaics 2, 303 (2012)

[Onno2022] A. Onno, C. Reich, S. Li, A. Danielson, W. Weigand, A. Bothwell, S. Grover, J. Bailey, G. Xiong, D. Kuciauskas, W. Sampath, and Z. C. Holman, Understanding What Limits the Voltage of Polycrystalline CdSeTe Solar Cells, Nat. Energy, 7, 400-408 (2022)

[Ross1967] R. T. Ross, Some Thermodynamics of Photochemical Systems, The Journal of Chemical Physics 46, 4590 (1967)

[Ruhle2016] S. Rühle, Tabulated values of the Shockley-Queisser limit for single junction solar cells, Solar Energy, 130, 139-147 (2016)

[Scarpulla2023] M. A. Scarpulla, B. McCandless, A. B. Phillips et al., CdTe-based thin film photovoltaics: Recent advances, current challenges and future prospects, Solar Energy Materials and Solar Cells, 255, 112289 (2023)

We agree with the reviewer’s point regarding additional sample analyses and have now added the following to the main text:

“We further explore our photoluminescence maps in Supporting Information Note 11, where we extract the local temperature, radiative and implied open circuit voltages from our data. We find that all samples have uniform temperature distributions slightly above room temperature, which is reasonable as we are using a focused laser beam to excite the sample. Our open circuit voltage maps further support our conclusions drawn here, again showing preferential passivation of grain boundaries for the 200 nm CdSe sample.”

and the Supporting Information Note reads:

“Supporting Information Note 11 – further map analyses

1. Temperature fitting

As $PL(E) \propto a(E)E^2 e^{-\frac{E}{k_B T}}$ (see supporting information note 7), where $a(E)$ is the sample absorption, and $a(E)$ is approximately constant well above the bandgap, we can state that in this energy region bandgap $\ln(PL(E)) = A - \frac{E}{k_B T}$, where A is a constant. We fitted our PL maps to extract the local temperature for each sample, which we present in Figure S16 for all samples. We find that all samples are at approximately 322 K during measurements except 200 nm CdSe, which is reasonable as we are using an intense laser beam to access the low PLQEs. We find 200 nm CdSe samples are approximately 4 K hotter, which we attribute to differences in the thermal diffusion coefficient with increased CdSe.

Figure S16. a)/b)/c)/d) Fitted temperature maps for 0 nm/50 nm/100 nm/200 nm CdSe underlayer.

2. Ideal radiative voltage

Based on the data presented in Figure S4 we can assume that $a(E) \sim 0.9$ well above the bandgap. Furthermore, in an ideal system (i.e. with no sub-bandgap PL), we assume the absorption follows an Urbach fit below the bandgap. Using these assumptions, and noting $PL(E) \propto a(E)E^2 e^{-\frac{E}{k_B T}}$, we extracted $a(E)$ from every point measured on our map (using the temperatures extracted above). We can therefore calculate the Shockley-Queisser radiative open circuit voltage noting

$$V_{OC,rad} = k_B T \ln \left(\frac{J_{sc}}{J_r} + 1 \right)$$

where $J_{sc} = q \int \phi_{AM1.5}(E) a(E) dE$ and $J_r = q \pi \int \phi_{bb}(E) a(E) dE$, where $\phi_{AM1.5}$ is the AM1.5 solar flux (per unit energy, per unit area) and ϕ_{bb} is the black body flux (per unit energy, per unit area, per unit solid angle) [8]. We present $V_{OC,rad}$ for our PL measurements in Figure S17, for solar cells operating at 300 K. As expected, $V_{OC,rad}$ falls for samples with more Se present, and there is relatively small variation of $V_{OC,rad}$ across the surface of each sample, suggesting relatively uniform samples from an absorption perspective.

Figure S17. a)/b)/c)/d) Radiative limit open circuit voltage maps for 0 nm/50 nm/100 nm/200 nm CdSe underlayer. Note colourbar is a different scale for each plot.

3. Implied open circuit voltage

As noted in the main text, an implied open circuit voltage can also be calculated from measured PLQY values through $V_{OC} = V_{OC,ideal} + \frac{k_B T}{q} \ln(PLQY)$ [8]. Here we take the PLQY of the main luminescence peak (removing any effects of sub-bandgap luminescence) to calculate the maximum open-circuit-voltage achievable with measured films, which we present in Figure S18, again for solar cells operating at 300 K. V_{OC} is lower in grain interiors for 200 nm CdSe, despite the higher bandgaps present in these regions.

Figure S18. a)/b)/c)/d) Implied open circuit voltage maps for 0 nm/50 nm/100 nm/200 nm CdSe underlayer, neglecting any effect of the sub-bandgap defect luminescence.

4. Average values

Here we present average values from the above analyses.

CdSe thickness (nm)	Temperature (K)	$V_{OC,ideal}$ (V)	V_{OC} (V)
0	322 ± 1	1.2183 ± 0.0006	0.8070 ± 0.0032
50	322 ± 1	1.2121 ± 0.0004	0.8167 ± 0.0043
100	321 ± 1	1.1974 ± 0.0007	0.8062 ± 0.0025
200	326 ± 1	1.1772 ± 0.0018	0.7895 ± 0.0024

Table S3. Average results and standard deviation for fits in this supplemental note."

[8] R. T. Ross, Some Thermodynamics of Photochemical Systems, The Journal of Chemical Physics 46, 4590 (1967).

Our SIMS results have allowed us to explain the manual shift observed in Figure S9 (now S10) with the supporting information reading as follows (noting the section above discusses the temperature in detail):

“Supporting Information Note 7 – prediction of photoluminescence from absorption measurements

We applied the van Roosbroeck-Shockley relation [7] to predict the spectral shape of the photoluminescence from our absorption measurements. Specifically, this relation states that $PL(E) \propto a(E)E^2 e^{-\frac{E}{k_B T}}$, where $a(E)$ is the measured absorption as a function of energy E and $k_B T$ the thermal energy, with the temperatures extracted in Supporting Information Note 11 used. We used the scattering subtracted, spatially averaged $a(E)$ (i.e. that presented in Figure S5) and found we had to shift predicted values up by 7 meV (i.e. predicted PL peaks were at lower energies than measured PL peaks) to obtain good agreement with spatially averaged measured photoluminescence. We attribute this shift to our PL measurements probing the front half of the CdTe layer (see main text for discussion) while absorption measurements probe the entire CdTe layer. As our SIMS results showed (supporting information note 2), near the back of the film is a small region with increased Se content, and thus marginally lower bandgap observed in absorption. Finally, we note that there is significant error in our below bandgap $a(E)$ measurements, so we only predict the energies close to the PL peak on the lower energy side. A comparison between experiment and theory is presented in Figure S10, with the good agreement implying that there is minimal Stokes shift prior to photoluminescence.

Figure S10. Comparison between measured (solid lines) and absorption predicted (dashed lines) photoluminescence for Cl-treated samples. Predicted PL was shifted by 7 meV to have good agreement (see text above).”

[7] W. van Roosbroeck and W. Shockley, Photon-Radiative Recombination of Electrons and Holes in Germanium, *Physical Review* 94, 1558 (1954).

Finally, we emphasise that while the absorbers we have made are of lower quality than world record samples, primarily our study does not focus on the quality of samples, but the change to their quality as

different quantities of Se are introduced. Our conclusions are related to the change in film quality rather than the absolute film quality.

3) Spectral range of PL measurements need to extend to lower energies than used in this study. E.g., as reported by Neupane 2023, Kuciauskas 2023, CdSeTe can have emission down to 0.8 eV (1600 nm), and such defect states can severely impact electronic properties. Data for increased Se compositions (which are still too low!) in Fig. 1 (200 nm CdSe), Fig. S8, Fig. S10 strongly suggest that measurements with InGaAs detectors (or similar) are also needed. Until that measurement is used, the topic in the manuscript title is not addressed.

We have presented data in the main text where we are fully confident of the absolute spectral calibration, which is to 950 nm. In the supporting information we now show data without spectral calibration to 1050 nm, the limit of a silicon spectrometer. We have added the following to the text:

“More surprisingly, as the quantity of Se is increased in Cl-treated samples, a longer, sub-bandgap wavelength (>900 nm) signal increases in relative intensity. This can be seen more clearly in Supporting Information Note 8, where we present data on a logarithmic scale (to 950 nm) and uncalibrated data to 1050 nm (noting we were only confident in our radiometric calibration to 950 nm).”

and the additional supporting information figure is as follows:

“

Figure S12. Uncalibrated average PL data for three measurements of each sample stretching to beyond 1000 nm, on a logarithmic scale. We note that we were confident in our absolute calibration only to 950 nm.”

Finally, we agree that measuring to significantly longer wavelengths using an InGaAs detector would be ideal. However, we do not have access to InGaAs detectors linked to state-of-the-art Raman microscopes with short wavelength (488 nm) excitation: typically InGaAs detectors would be used with Raman microscopes with 1050 nm or longer excitation, which is not useful for this study. For comparison we note that we are not aware of any papers on either halide perovskite or CdTe samples that use InGaAs cameras to map luminescence at the microscale. We have added the following to the main text on this point:

“Future work to explore this effect should explore longer wavelength photoluminescence, potentially using a Raman microscope with an InGaAs detectors (that can access wavelengths longer than 1050 nm) coupled to short wavelength excitation (noting Raman microscopes with InGaAs detectors currently use long wavelength, >900 nm, excitation lasers).”

4) There are additional more specific questions about the data and analysis, but they are not needed at this stage.

We would be delighted to answer any additional questions that the reviewer has.

Reviewer #3 (Remarks to the Author):

1. The intrinsic physical mechanism of the impact of the introduction of Se on the photoelectric properties of CdSeTe thin films is very important for the development of high-efficiency CdTe thin film solar cells. The author used micro-area PL and other technologies to study the impact of Se introduction in CdSeTe on its properties. No similar research has been reported in the literature so far. This paper is of great reference for researchers working on CdTe thin film solar cells and is worthy of publication.

We are delighted that the reviewer considers our quality reasonable for publication in Nature Communications.

2. However, in order to ensure that the short-circuit current density of the device is sufficiently high, when preparing CdTe solar cells with an absorption layer including CdSeTe, specific processes are usually adopted to control the Se composition to obtain a CdSeTe layer with a bandgap of ~1.39 eV. In fact, the QE curve of the CdTe solar cell with the highest photoelectric conversion efficiency reported in the literature shows that its long wavelength cutoff edge extends to 1.39 eV [Green MA, Dunlop ED, Yoshita M, et al. Solar cell efficiency tables (Version 63). Prog Photovolt Res Appl. 2023;1-11. doi:10.1002/pip.3750.]. However, the band gap of the sample studied in this article is only ~1.44 eV. In order to better understand the intrinsic mechanism by which the introduction of Se affects the properties of CdSeTe films, it is recommended that the authors provide characterization results of samples with bandgaps ranging from 1.45 eV to 1.39 eV.

As this is one of the first study of samples on glass we have kept our synthesis method as similar as possible to previous literature studies on devices, where CdSe thicknesses greater than 200 nm CdSe have resulted in severely reduced quantum efficiencies, postulated to be caused by unconsumed wurtzite CdSe [Poplawsky2016, Baines2018]. The bandgap of our samples ranges from 1.49 eV to 1.44 eV (from Tauc fitting) and, unfortunately, samples with higher thicknesses of CdSe would have been even more prone to delamination. We have added the following points to the text:

“50 nm, 100 nm and 200 nm CdSe layers were deposited on the substrates by radio-frequency sputtering with a power density of 1.32 W/cm², Ar pressure of 5 mTorr and a substrate temperature of 200°C. These thicknesses were selected based on previous reports of CdSe layers thicker than 200 nm leading to unconsumed, photoinactive CdSe in devices [Poplawsky2016, Baines2018].”

[Poplawsky2016] J. D. Poplawsky, W. Guo, N. Paudel, A. Ng, K. More, D. Leonard and Y. Yan, Structural and Compositional Dependence of the CdTe_xTe_{1-x} Alloy Layer Photoactivity in CdTe-Based Solar Cells, Nature Communications, 7(1), 12537 (2016)

[Baines2018] T. Baines, Z. Guillaume, L. Bowen, T. P. Shalvey, S. Mariotti, K. Durose and J. D. Major, Incorporation of CdSe Layers into CdTe Thin Film Solar Cells, Solar Energy Materials and Solar Cells, 180, 196-204 (2018)

3. In addition, in order to help readers better understand and reproduce the experimental results of the manuscript, the author should supplement more details on the preparation process and characterization techniques of certain samples, such as:

(1) The relevant details of the "Photoluminescence and Photoluminescence Quantum Yield (PLQY)" test, such as the size or spatial resolution of the excited light spot, as well as the control mechanism of the movable sample stage used and the spatial resolution of the movable sample stage.

The Methods section now reads:

“Photoluminescence and photoluminescence quantum yield (PLQY): was carried out on a Renishaw inVia Raman Microscope RE04 using a Coherent sapphire 488 SF NX excitation, incident on the air-CdSe_xTe_{1-x} interface as the thickness of the glass substrate prevented good focus on the glass-CdSe_xTe_{1-x} interface. All maps were recorded via snake scanning of the diffraction limited spot across the sample using an MS20 stage with minimum step size of 100 nm, dynamic trajectory control via a high speed

microprocessor, via serial communication, and real time positional tracking using linear encoder feedback.”

(2)Details related to the "Microscale transmission and reflection measurements" test, such as the size of the spot after the incident light is focused on the surface of the sample, and whether a sample moving table was also used during the testing process? The manuscript mentions "due to spatial and noise resolution limits in our measurements, we were unable to draw definitive conclusions from absorption measurements all". Please provide a detailed explanation.

The methods section now reads:

“The area illuminated was $> 150 \mu\text{m}$ in all cases and transmission/reflection measurements were performed in a subregion within the illuminated spot (this avoids issues with inhomogeneous camera heating). Measurements were taken simultaneously over a lineslice by closing the slit at the entrance of the spectrometer to isolate the region of interest. The spectrometer camera, which was aligned at a conjugate focal plane, thus recorded the spectrum versus position in the region of interest without the need for a moving stage. In all measurements, transmitted or reflected light was collected by a 60x Nikon S Plan Fluor objective.”

In the main text we state:

“However, due to the strong variation in below bandgap scattering across the sample (Figure 1c) it was difficult to draw a full conclusion, due to the error associated with the Tauc fitting (see Supporting Information Note 5 for more detail).”

and Supporting Information Note 5 states:

“However, subtraction of below-bandgap scattering prior to Tauc fitting introduces some noise in these measurements (as can be seen in Figure S7, point to point variation is around 0.01 eV), preventing us from definitively stating that bandgaps are always lower at grain boundaries.”

4. The sample preparation technique used in this manuscript may result in a gradually decreasing distribution of Se content from the substrate to the interface between the film and air. When conducting PL testing, the measured fluorescence intensity is related to the absorption length of the excitation light wavelength used (usually the reciprocal of the absorption coefficient), which means that the fluorescence intensity is a contribution of a certain depth range of CdSeTe samples with non-uniform Se distribution. The manuscript should analyze whether the vertical distribution of Se affects its conclusion. If possible, the manuscript should provide characterization results of the longitudinal Se distribution of the sample.

We thank the reviewer for highlighting this important point which we agree was lacking in the text. We have now carried out secondary ion mass spectrometry (SIMS) measurements to characterise the vertical distribution of Se and Te within the sample, with this text reading as follows:

“We carried out a number of morphological measurements to better understand the quality of the fabricated films, with results presented in Supporting Information note 2. Firstly, we carried out secondary ion mass spectrometry (SIMS) on samples with 50 nm and 200 nm CdSe underlayer to better understand the vertical distribution of Se throughout the films. In contrast to devices [Artegiani2021], we find that the ratio of Se to Te is approximately constant through the majority of the film, with a $< 0.5 \mu\text{m}$ region close to the glass substrate containing a small increase in Se content, making these films close to ideal for spectroscopic analyses, as is presented in Figure 1b as a function of SIMS etch number (see Supporting Information note 2 for further details). These measurements also confirmed that thicker CdSe underlayers resulted in greater Se content throughout the films. Secondly, we used interferometry measurements to record the thickness of our films, which we found to be between $2 \mu\text{m}$ and $2.5 \mu\text{m}$ in all cases. Finally, we used atomic force microscopy to demonstrate that in non-delaminated regions,

similar grain morphologies and surface roughnesses were observed for all chlorine-treated samples. We note for these measurements, and all others presented in the paper, multiple positions were recorded on measured samples with representative results being presented in the main text and supporting information.

Figure 1. a) white light reflection image of a Cl-treated 0 nm CdSe sample. Extended dark areas correspond to region of delamination. Overlaid on this plot is a lineslice along which we measured sample absorption. b) Ratio of Se to Te content as a function of etch number, as recorded via secondary ion mass spectrometry. Etch number 200 approximately corresponds to the glass substrate at the back of the film and each line to a different measurement (see legend). c) absorption + scatter results obtained for a 0 nm CdSe, Cl-treated sample. Above the main plot is the white light image of the sample with a blue dashed line highlighting the region measured. d) change in bandgap with CdSe thickness, via Tauc fitting of average absorption spectra from microscopic measurements, for samples both with and without a prior Cl-treatment. e) absorption + scatter for specific distances, as marked with colour arrows for specific positions in c), as the measurement moves across a grain. Weaker below-bandgap scattering is seen when measuring on a grain (corresponding to 51.2 μm and 52.0 μm).”

[Artegiani2021] E. Artigiani, P. Punathil, V. Kumar, M. Bertocello, M. Meneghini, A. Gasparotto, and A. Romeo, Effects of CdTe Selenization on the Electrical Properties of the Absorber for the Fabrication of $\text{CdSe}_x\text{Te}_{1-x}/\text{CdTe}$ Based Solar Cells, *Solar Energy* 227, 8 (2021).

And the following text to the supporting information:

“SIMS was used to record the vertical distribution of Se and Te throughout films. Specifically, we studied samples fabricated with 50 nm CdSe and 200 nm CdSe underlayers (noting the time intensive nature of this measurement prevented us from also exploring 100 nm CdSe samples). We recorded the intensity of Se and Te present in the film, then used a Cs^+ ion beam to remove a thin layer of the film

repeatedly (see methods), building a vertical profile. This process was repeated until we reached the substrate, following previous analyses of CdSeTe samples [5]. We present our results in Figure S1, for three regions measured in the 50 nm CdSe sample and two regions measured in the 200 nm CdSe sample, as a function of the etch number in the measurement. The absolute intensities of the Se and Te signals were found to vary between samples, as is presented in Figure S1a and S1b, as is expected for samples with different surfaces exposed and small variations in measurement parameters between each experiment. However, the key quantity is the ratio of Se to Te, which we present in Figure S1c. This reveals three important points: i) the Se/Te ratio is relatively consistent across different sample regions; ii) samples fabricated with 200 nm CdSe underlayers have significantly higher Se relative to Te; and iii) within each sample the ratio of Se to Te is relatively uniform throughout the film, with significant changes occurring only close to 200 etches i.e. extremely close to the glass substrate (see Figure S1a/b). This shows that the vast majority of our films have relatively uniform proportions of Se and Te and that ratio can be controlled via the thickness of the initial CdSe layer.

Figure S1. SIMS intensity for a) Se and b) Te as a function of Etch number (with 0 etches being the air-exposed surface). c) ratio of Se to Te intensities. Legend in c) applies to all plots.”

[5] E. Artegiani, P. Punathil, V. Kumar, M. Bertoncello, M. Meneghini, A. Gasparotto, and A. Romeo, Effects of CdTe Selenization on the Electrical Properties of the Absorber for the Fabrication of Cd_{Se}xTe_{1-x}/CdTe Based Solar Cells, *Solar Energy* 227, 8 (2021).

Finally, we have added a discussion of exactly the region of the samples we are probing with PLQY:

“Our PLQY values also allow us to consider where within our film our luminescence measurements probe. Specifically, at our excitation wavelength of 488 nm CdTe’s absorption depth is 88 nm [Treharne2011]. Charge diffusion lengths of up to ~ 30 μm have been observed in single crystals of CdTe [Ščajev2022], and as noted above the best PLQY results reported are on the order of 10-2 [Kuciauskas2020]. As our PLQY values of samples on glass are at least four orders of magnitude lower, and PLQY is (approximately) inversely proportional to charge lifetime, an upper bound on charge diffusion lengths in our films is 1 μm. Therefore, as the front half of our films have a uniform Se to Te ratio (Supporting Information Note 2), our photoluminescence signals originate from a region where the ratio of Se to Te is constant.”

[Kuciauskas2020] D. Kuciauskas, J. Moseley, P. Ščajev, and D. Albin, Radiative Efficiency and Charge-Carrier Lifetimes and Diffusion Length in Polycrystalline CdSeTe Heterostructures, *Phys. Status Solidi RRL*, 14(3), 1900606 (2020)

[Ščajev2022] Ščajev, P., Mekys, A., Subačius, L. et al. Impact of dopant-induced band tails on optical spectra, charge carrier transport, and dynamics in single-crystal CdTe. *Scientific Reports*, 12, 12851 (2022).

[Treharne2011] R. E. Treharne, A. Seymour-Pierce, K. Durose, K. Hutchings, S. Roncallo and D. Lane. Optical Design and Fabrication of Fully Sputtered CdTe/CdS Solar Cells. *Journal of Physics: Conference Series*. 286 012038 (2011)

5. Usually, it is challenging to obtain the thin and dense CdTe films with using CSS technology. The manuscript should provide more details concerning CSS process and the thickness of the prepared samples.

Details of the CSS fabrication process used are given in the Methods section. The following has been added, but if the reviewer feels any specific details are lacking we would be happy to provide them:

“CdTe was deposited by close-space sublimation (CSS), at a deposition pressure of 5 Torr in an O_2 atmosphere, with a source temperature of 600°C and a substrate temperature of 550°C. A 20 min post-growth anneal was performed in the CSS chamber immediately after deposition, at a substrate temperature of 550°C and a pressure of 400 Torr. Film thicknesses were found to be between 2 μm and 2.5 μm from interferometry measurements.”

Reviewer #4/5 (Remarks to the Author):

The stack is glass/CdSe/CdTe. There is no passivation layer in the front or the back leading to a high degree of recombination in the front and back surfaces. This makes it difficult to draw conclusions about passivation in the bulk/grain boundaries.

We have specifically avoided using a passivation layer, to investigate the role that Se directly has on samples without other competing effects, which we now discuss thoroughly in Supporting Information Note 1:

“Supporting Information Note 1 – luminescence with charge extraction layers present

We study samples on glass to remove charge extraction effects from our spectroscopic results. Here we briefly discuss additional effects that can be observed with charge extraction layers or at interfaces between two electronically active materials, noting that this is an area of ongoing spectroscopic research. Specifically, we give two examples from complementary fields.

Our first example is time resolved photoluminescence (TRPL) from halide perovskites. Several studies have focused on changes to TRPL signals when a halide perovskite is on glass compared to on a charge extraction layer, including Stolterfoht et al [1]. Here it was observed that a halide perovskite had significantly shorter TRPL lifetime when placed on a charge extraction layer (see figure 3 in this paper), noting that shorter TRPL lifetimes are equivalent to weaker luminescence signals in steady state decays. The authors attribute this reduced lifetime to a mixture of charge extraction and interfacial recombination. Importantly, while TRPL lifetimes are significantly longer following interfacial passivation (see figure 5c and 5f in this paper), there is still a rapid initial drop and charge lifetime is still lower than on glass. This shows that a reduction in TRPL lifetime (i.e. luminescence) can be due to charge extraction or interfacial charge traps, and it is difficult to deconvolute these competing effects. This has been further explored by a number of others in this field [2,3], generally demonstrating that charge transport layers can have a multitude of effects on the strength of luminescence signals. Notably, significant changes are seen even with a single transport layer deposited on the active layer, which is a form of open circuit (likely due to interfacial recombination and some charges being extracted to the transport layer and then subsequently undergoing non-radiative recombination across the interface).

Our second example is from organic semiconductors. Here it is common to study heterojunctions between two electronically active materials. Below-bandgap photoluminescence peaks have been observed due to charge transfer states between two organic semiconductors, for example as shown by Ng et al [4]. This demonstrates that when multiple electronically active materials are present their interaction can introduce interfacial luminescence states. By studying bare CdTe samples on glass we are able to rule out that the below-bandgap luminescence states seen in our system originate from a charge transfer state or similar.

Both these examples demonstrate the importance of studying samples on glass (or other inert substrate) when focusing on the effect of a single material within a solar cell device stack.”

[1] M. Stolterfoht et al., Visualization and Suppression of Interfacial Recombination for High-Efficiency Large-Area Pin Perovskite Solar Cells, *Nat Energy* 3, 847 (2018).

[2] L. Krückemeier, Z. Liu, B. Krogmeier, U. Rau, and T. Kirchartz, Consistent Interpretation of Electrical and Optical Transients in Halide Perovskite Layers and Solar Cells, *Adv. Energy Mater.* 11, 2102290 (2021).

[3] L. Krückemeier, B. Krogmeier, Z. Liu, U. Rau, and T. Kirchartz, Understanding Transient Photoluminescence in Halide Perovskite Layer Stacks and Solar Cells, *Advanced Energy Materials* 11, 2003489 (2021).

[4] A. M. C. Ng, A. B. Djurišić, W.-K. Chan, and J.-M. Nunzi, Near Infrared Emission in Rubrene:Fullerene Heterojunction Devices, *Chemical Physics Letters* 474, 141 (2009).

Due to low PLQYs we believe that the majority of signal originates from close to where the sample was excited, whereas longer carrier lifetimes increase ambiguity in where in the sample signal was collected from. Therefore, as we are exciting some regions that have grain boundaries and others at grain interiors, our measurements directly explore effects between grain boundaries and grain interiors. Thus we believe in our measurements it is possible to differentiate between grain boundaries and grain interiors. We have added the following line to the text to clarify this point:

“Our PLQY values also allow us to consider where within our film our luminescence measurements probe. Specifically, at our excitation wavelength of 488 nm CdTe’s absorption depth is 88 nm [Treharne2011]. Charge diffusion lengths of up to $\sim 30 \mu\text{m}$ have been observed in single crystals of CdTe [Ščajev2022], and as noted above the best PLQY results reported are on the order of 10^{-2} [Kuciauskas2020]. As our PLQY values of samples on glass are at least four orders of magnitude lower, and PLQY is (approximately) inversely proportional to charge lifetime, an upper bound on charge diffusion lengths in our films is $1 \mu\text{m}$. Therefore, as the front half of our films have a uniform Se to Te ratio (Supporting Information Note 2), our photoluminescence signals originate from a region where the ratio of Se to Te is constant.”

[Kuciauskas2020] D. Kuciauskas, J. Moseley, P. Ščajev, and D. Albin, Radiative Efficiency and Charge-Carrier Lifetimes and Diffusion Length in Polycrystalline CdSeTe Heterostructures, *Phys. Status Solidi RRL*, 14(3), 1900606 (2020)

[Ščajev2022] Ščajev, P., Mekys, A., Subačius, L. et al. Impact of dopant-induced band tails on optical spectra, charge carrier transport, and dynamics in single-crystal CdTe. *Scientific Reports*, 12, 12851 (2022).

[Treharne2011] R. E. Treharne, A. Seymour-Pierce, K. Durose, K. Hutchings, S. Roncallo and D. Lane. Optical Design and Fabrication of Fully Sputtered CdTe/CdS Solar Cells. *Journal of Physics: Conference Series*. 286 012038 (2011)

Finally, we note that our study specifically focuses on the effects of different quantities of CdSe i.e. we focus on changes in film quality with additional CdSe. These differential changes are evident independent of the absolute PLQY values.

The CdCl₂ treatment led to delamination and the treatment was not optimum leaving defects in the film and difficult to draw conclusions about the nature of PL. Also, the CdCl₂ treatment did not change the bandgap indicating very low intermixing.

Firstly, we highlight to the reviewer that a MgCl₂ treatment was used, not CdCl₂. Unfortunately, due to the samples’ deposition on glass, there was an unavoidable trade-off between the Cl treatment’s passivation efficacy and the extent of delamination, so in that sense the Cl treatment used was “optimal” within the constraints of the samples. We highlight that all spectroscopic measurements were spatially resolved, and we avoided all areas where delamination was present, as is highlighted in the following in the main text:

“Due to the spatially resolved nature of the measurements used, analysis was able to focus solely on areas where grains remain intact.”

Whilst the Cl treatment did not affect the absorption measurements, it did significantly improve PLQEs in all samples and cause a shift in the PL emission peak for samples with a CdSe layer. Combined with SIMS measurements of the Se profile below, this indicates that the majority of CdSe/CdTe interdiffusion occurred prior to the Cl treatment, but the Cl treatment was necessary for efficient luminescence from the CdSe_xTe_{1-x} phase. Since the same Cl treatment conditions were used for all Cl treated samples, we are confident that the trends observed in the photophysical properties are due to changes in the Se content and not due to Cl treatment effects. The following has been added to the text to further clarify the reasons for the choice of Cl treatment conditions:

“Various Cl treatment conditions were tested, with the approach used in this work providing the optimal balance between Cl treatment passivation and reduced sample delamination.”

Due to the low CdCl₂ treatment the redistribution of Se is likely to be limited and high non-uniform Se concentration and making it difficult to study the effects of Se.

We thank the reviewer for this comment. We have now carried out secondary ion mass spectrometry (SIMS) measurements to study the distribution of Se and Te from the front to the back of the sample. Importantly, in the region studied by our luminescence measurements we find that the ratio of Se to Te is approximately constant. We have added the following to the text to clarify this point:

“We carried out a number of morphological measurements to better understand the quality of the fabricated films, with results presented in Supporting Information note 2. Firstly, we carried out secondary ion mass spectrometry (SIMS) on samples with 50 nm and 200 nm CdSe underlayer to better understand the vertical distribution of Se throughout the films. In contrast to devices [Artegianni2021], we find that the ratio of Se to Te is approximately constant through the majority of the film, with a < 0.5 μm region close to the glass substrate containing a small increase in Se content, making these films close to ideal for spectroscopic analyses, as is presented in Figure 1b as a function of SIMS etch number (see Supporting Information note 2 for further details). These measurements also confirmed that thicker CdSe underlayers resulted in greater Se content throughout the films. Secondly, we used interferometry measurements to record the thickness of our films, which we found to be between 2 μm and 2.5 μm in all cases. Finally, we used atomic force microscopy to demonstrate that in non-delaminated regions, similar grain morphologies and surface roughnesses were observed for all chlorine-treated samples. We note for these measurements, and all others presented in the paper, multiple positions were recorded on measured samples with representative results being presented in the main text and supporting information.

Figure 1. a) white light reflection image of a Cl-treated 0 nm CdSe sample. Extended dark areas correspond to region of delamination. Overlaid on this plot is a lineslice along which we measured sample absorption. b) Ratio of Se to Te content as a function of etch number, as recorded via secondary ion mass spectrometry. Etch number 200 approximately corresponds to the glass substrate at the back of the film and each line to a different measurement (see legend). c) absorption + scatter results obtained for a 0 nm CdSe, Cl-treated sample. Above the main plot is the white light image of the sample with a blue dashed line highlighting the region measured. d) change in bandgap with CdSe thickness, via Tauc fitting of average absorption spectra from microscopic measurements, for samples both with and without a prior Cl-treatment. e) absorption + scatter for specific distances, as marked with colour arrows for specific positions in c), as the measurement moves across a grain. Weaker below-bandgap scattering is seen when measuring on a grain (corresponding to 51.2 μm and 52.0 μm).

[Artegianni2021] E. Artegianni, P. Punathil, V. Kumar, M. Bertocello, M. Meneghini, A. Gasparotto, and A. Romeo, Effects of CdTe Selenization on the Electrical Properties of the Absorber for the Fabrication of $\text{CdSe}_x\text{Te}_{1-x}/\text{CdTe}$ Based Solar Cells, *Solar Energy* 227, 8 (2021).

And the following text to the supporting information:

“SIMS was used to record the vertical distribution of Se and Te throughout films. Specifically, we studied samples fabricated with 50 nm CdSe and 200 nm CdSe underlayers (noting the time intensive nature of this measurement prevented us from also exploring 100 nm CdSe samples). We recorded the intensity of Se and Te present in the film, then used a Cs^+ ion beam to remove a thin layer of the film repeatedly (see methods), building a vertical profile. This process was repeated until we reached the substrate, following previous analyses of CdSeTe samples [5]. We present our results in Figure S1, for three regions measured in the 50 nm CdSe sample and two regions measured in the 200 nm CdSe sample, as a function of the etch number in the measurement. The absolute intensities of the Se and Te signals were found to vary between samples, as is presented in Figure S1a and S1b, as is expected for samples with different surfaces exposed and small variations in measurement parameters between each experiment. However, the key quantity is the ratio of Se to Te, which we present in Figure S1c. This reveals three important points: i) the Se/Te ratio is relatively consistent across different sample regions; ii) samples fabricated with 200 nm CdSe underlayers have significantly higher Se relative to Te; and iii) within each sample the ratio of Se to Te is relatively uniform throughout the film, with significant changes occurring only close to 200 etches i.e. extremely close to the glass substrate (see Figure S1a/b). This shows that the vast majority of our films have relatively uniform proportions of Se and Te and that ratio can be controlled via the thickness of the initial CdSe layer.

Figure S1. SIMS intensity for a) Se and b) Te as a function of Etch number (with 0 etches being the air-exposed surface). c) ratio of Se to Te intensities. Legend in c) applies to all plots.”

[5] E. Artegianni, P. Punathil, V. Kumar, M. Bertocello, M. Meneghini, A. Gasparotto, and A. Romeo, Effects of CdTe Selenization on the Electrical Properties of the Absorber for the Fabrication of $\text{CdSe}_x\text{Te}_{1-x}/\text{CdTe}$ Based Solar Cells, *Solar Energy* 227, 8 (2021).

Finally, we have added a discussion of exactly the region of the samples we are probing with PLQY:

“Our PLQY values also allow us to consider where within our film our luminescence measurements probe. Specifically, at our excitation wavelength of 488 nm CdTe’s absorption depth is 88 nm [Treharne2011]. Charge diffusion lengths of up to $\sim 30 \mu\text{m}$ have been observed in single crystals of CdTe [Ščajev2022], and as noted above the best PLQY results reported are on the order of 10⁻² [Kuciauskas2020]. As our PLQY values of samples on glass are at least four orders of magnitude lower, and PLQY is (approximately) inversely proportional to charge lifetime, an upper bound on charge diffusion lengths in our films is 1 μm . Therefore, as the front half of our films have a uniform Se to Te ratio (Supporting Information Note 2), our photoluminescence signals originate from a region where the ratio of Se to Te is constant.”

[Kuciauskas2020] D. Kuciauskas, J. Moseley, P. Ščajev, and D. Albin, Radiative Efficiency and Charge-Carrier Lifetimes and Diffusion Length in Polycrystalline CdSeTe Heterostructures, *Phys. Status Solidi RRL*, 14(3), 1900606 (2020)

[Ščajev2022] Ščajev, P., Mekys, A., Subačius, L. et al. Impact of dopant-induced band tails on optical spectra, charge carrier transport, and dynamics in single-crystal CdTe. *Scientific Reports*, 12, 12851 (2022).

[Treharne2011] R. E. Treharne, A. Seymour-Pierce, K. Durose, K. Hutchings, S. Roncallo and D. Lane. Optical Design and Fabrication of Fully Sputtered CdTe/CdS Solar Cells. *Journal of Physics: Conference Series*. 286 012038 (2011)

As many reported earlier, Se diffuses through grain boundaries into the bulk. Figure 3 e) indicates poor intermixing of CdSe/CdTe with low Se concentration in the bulk in comparison to grain boundaries. Resulting in better PLQY in the grain boundaries than in the bulk.

Our results do not necessarily suggest poor intermixing of Se to the bulk of grains for a specific sample. However, the important point is the trend seen across samples independent of the degree of intermixing within a specific sample – when more Se is present, including in grain interiors, we see an increase in below bandgap luminescence from grain interiors. Therefore, the comparison between our samples is justifiable and gives new scientific information, which suggests that increase in Se in grain interiors can introduce traps in grain interiors. We have added the following to the text to clarify this point:

“We note that while our samples may not have optimal Se intermixing, the trend we see is consistent with increasing Se content across several regions measured, meaning the correlation between below-bandgap traps and increasing Se content is justifiable.”

REVIEWER COMMENTS

Reviewer #1 (Remarks to the Author):

I thank the authors for their detailed responses. I have a few comments:

1) Fig 1a in the text shows the very inhomogeneous, partly delaminated CdSeTe film with large islands where there are no CdSeTe grains present. The SIMS measurements on the delaminated film have a spot size of 80 microns (see methods). This means that during the SIMS depth profiles, when at low etch numbers, which are supposed to be sampling from the top layers of the CdSeTe only, the Cs⁺ beam will also be sampling from delaminated and partially delaminated regions, skewing the depth profile data. It is therefore very hard to trust the SIMS data. The authors do take measurements from multiple regions, but by the looks of the film, the issue is going to be the same all over the substrate (and how did they choose which profiles were 'representative' and therefore which to include in the figure?). The authors need to adequately address this issue.

2) In my previous comment I said: 'The Bg of the '0 nm CdSe' film is only 0.04 eV higher than the '200 nm CdSe' film. This suggests that there is a relatively small amount of Se present in the films (~5 at.% Se even in their most Se-rich film).

The authors responded with: 'Published bandgap bowing relationships for CdSe_xTe_{1-x} indicate the bandgap reduction of 0.05 eV observed in this work corresponds to a Se content of 10 at% [J. Yang and S.-H. Wei, Chin. Phys. B 28, 086106 (2019)].'

However, reviewing the bandgap bowing data from the cited paper Yang paper (Fig 2) reveals that an 0.05 eV Bg reduction corresponds to an ANION alloying fraction change of $x=0.1$ (i.e. 5 at.%, not 10 at% as stated).

Reviewer #2 (Remarks to the Author):

Revised manuscript by Bowman et al is significantly improved. It is a technically correct study, with interesting data on CdSe(0.1)Te(0.9) photoluminescence and other properties.

I will reiterate that results are not relevant for CdSeTe solar cells (please see below). If sentences 1-2 from the abstract and PV-related paragraphs from introduction are removed, results can be published, they will add to the literature on CdSe(0.1)Te(0.9) electronic properties.

Manuscript by Bowman is limited to $x < 0.1$ in CdSe(x)Te(1- x). But it has been shown that passivation effect for $x=0.1$ composition is insignificant. E.g., Amarasinghe et al (APL 118, 211102 (2021))

reported TRPL lifetimes as a function of x . These lifetimes were 30 ns for $x=0$, 60 ns for $x=0.1$, 800 ns for $x=0.2$, and 1300 ns for $x=0.3$. Thus, to understand passivation (stated in the title), higher x values must be studied.

Preparation of $\text{CdSe}_x\text{Te}_{1-x}$ films with variation in x is widely used, it is a standard approach in this field. E.g., very recent paper (Frouin et al, APL Mater. 12, 031135 (2024)) has reported CL data for $x=0, 0.1, 0.2, 0.3$, and 0.4 . Frouin et al show that radiative emission changes very substantially for higher x values (Figure 1d and SI). Spatially resolved CL data in Frouin et al is in good agreement with macroscopic PL data in [Kuciauskas2023], measured on samples fabricated at different institutions (NREL for Frouin and Colorado State for Kuciauskas), meaning that defect features in CdSeTe reproducible for absorbers from different labs. PV relevant compositions must be studied if the goal is to provide guidance to PV material development.

Second (smaller) point is about excitation conditions used in confocal PL: $5.9\text{E}6 \text{ mW/cm}^2$, or approximately 60,000 Suns. Solar cells are used at 1 Sun. Authors show that trapping regime still applies at 60,000 Suns, meaning that material studied by Bowman has a lot of defects. But it is not shown that defects are the same at 60,000 Suns and 1 Sun (e.g., it is likely that some trap states are filled at 60,000 Suns). Using high excitation in confocal PL is understandable, and data is good, but results don't directly apply at 1 Sun. E.g., it might be possible to measure macroscopic PL at 1 Sun, to compare confocal and standard PL emission data.

In summary, a better, more comprehensive paper on this topic was recently published. Bowman et al manuscript can be published as well, as a confirmation of radiative emission mapping with a complimentary method (confocal PL vs CL), but on a more limited range of CdSeTe compositions.

Reviewer #3 (Remarks to the Author):

In this manuscript, the author aims to understand the influence of Se introduction on the properties of $\text{CdSe}_x\text{Te}_{1-x}$ films, which was formed by sequentially depositing CdSe and CdTe films and followed by high-temperature heat treatment in a chlorine containing atmosphere, through micro PLQY characterization technology and other characterization methods. As stated in the previous review comments, the research perspective proposed in the manuscript on this scientific issue has not been reported in relevant literature. It is expected that researchers in the field of CdTe solar cells will be very interested in the research results of this manuscript. So, this paper is well worth publishing.

However, a very important factor to consider when conducting this study is to clarify which scale range of $\text{CdSe}_x\text{Te}_{1-x}$ films correspond to the measured PLQY results, and whether the specific Se content and structure of $\text{CdSe}_x\text{Te}_{1-x}$ films in this scale range are clear. Therefore, in the previous review comments, it was suggested that the author conduct SIMS characterization to clarify the distribution of Se in $\text{CdSe}_x\text{Te}_{1-x}$ thin films. From the SIMS characterization results submitted by the author, it can be seen that the concentration of Se in the direction from the glass substrate to the air

side gradually decreases. This is consistent with the process of preparing the sample mentioned in the manuscript, which involves depositing a CdSe layer first, followed by depositing a CdTe layer, allowing Se to diffuse during the high-temperature process to obtain a specific CdSexTe1-x thin film sample. Using this process, it is possible to obtain CdSexTe1-x samples with uniformly distributed Se in the direction perpendicular to the substrate surface, but very harsh conditions may be required, such as very long time, appropriate high temperature, etc. Obviously, the SIMS characterization results provided in the revised manuscript indicate that the process used by the author to prepare the samples did not ensure the uniform longitudinal distribution of Se content in CdSexTe1-x thin film samples.

Furthermore, the authors clearly indicate in the sample testing section on page 12 of the revised manuscript that when performing PLQY testing, using 488nm excitation light, the absorption depth is only ~88nm. In the sample preparation section on page 13, it is also clearly stated that the sample thickness is 2~2.5 microns. According to literature, using a process similar to the one described in the manuscript, in which the samples were treated in a chlorine-containing atmosphere, the diffusion depth of Se in CdTe is usually ~1 micron [1, 2]. Combined with the SIMS characterization results provided by the manuscript, which reflects the distribution of Se elements perpendicular to the substrate direction, the PLQY value only comes from the contribution of less than 100nm depth range near the air side in CdSexTe1-x thin film samples, which is the region with the lowest Se content in CdSexTe1-x thin film samples. So, based on the current PLQY measurements in the revised manuscript and the characterization results of Se content in CdSexTe1-x thin film samples, it is scientifically insufficient and logically not rigorous to derive the rule of Se content's influence on the properties of CdSexTe1-x thin films.

In summary, the following suggestions are proposed:

- (1) Prepare CdSexTe1-x thin films with uniform longitudinal distribution of Se content using other techniques, and then perform PLQY characterization.
- (2) Alternatively, a technique similar to Ar ion etching in XPS can be used to sequentially etch CdSexTe1-x thin film samples, reducing their thickness and characterizing PLQY, in order to obtain the variation pattern of how the longitudinal Se content distribution affects the PLQY measurement results of the samples.
- (3) To increase the absorption depth of excitation light, an excitation light source with photon energy slightly higher than the bandgap width of CdSexTe1-x can be used in PLQY testing. But this not only requires a uniform longitudinal distribution of Se content in CdSexTe1-x films, but also requires that the thickness of CdSexTe1-x films is equivalent to the absorption depth, ensuring that the results of PLQY can reflect the chemical environment related to Se in the overall CdSexTe1-x films.

We have colour coded our response for clarity. Comments from reviewers are in **blue**, our responses are in **green** and any changes to the manuscript made are given in **black** text.

Reviewer #1 (Remarks to the Author):

I thank the authors for their detailed responses. I have a few comments:

1) Fig 1a in the text shows the very inhomogeneous, partly delaminated CdSeTe film with large islands where there are no CdSeTe grains present. The SIMS measurements on the delaminated film have a spot size of 80 microns (see methods). This means that during the SIMS depth profiles, when at low etch numbers, which are supposed to be sampling from the top layers of the CdSeTe only, the Cs+ beam will also be sampling from delaminated and partially delaminated regions, skewing the depth profile data. It is therefore very hard to trust the SIMS data. The authors do take measurements from multiple regions, but by the looks of the film, the issue is going to be the same all over the substrate (and how did they choose which profiles were ‘representative’ and therefore which to include in the figure?). The authors need to adequately address this issue.

We thank the reviewer for highlighting this important point. Whilst we understand the concern here, delamination should not influence the SIMS analysis: delamination occurs between the CdSeTe film and the glass interface, rather than part-way through the film, and that delamination resulted in lifting of the CdSeTe film from the substrate rather than complete removal. Dark spots appearing on images are due to the focal position of the lens rather than completely blank spaces in the material. Therefore, the profile from the front to the back of the film is the same for both laminated and delaminated regions and the sputtering rate for both regions will be the same. If this were not the case an approximately constant Se/Te ratio would only result from the Se/Te ratio being constant from both these regions, or a very unlikely balancing of Se to Te ratio between these regions. We carried out SIMS at several points across the sample, with consistent results from all measurements. We have added the following to the methods to highlight the reviewer’s point:

“While our SIMS measurements record regions with both laminated and delaminated sample, we note delamination occurs between the thin film and the glass rather than within the sample, meaning the Se/Te distribution should be the same in both regions. Additionally, multiple measurements were taken at different points across samples to verify the Se distribution was consistent and that measurements were not unduly influenced by any delamination.”

2) In my previous comment I said: ‘The Bg of the ‘0 nm CdSe’ film is only 0.04 eV higher than the ‘200 nm CdSe’ film. This suggests that there is a relatively small amount of Se present in the films (~5

at.% Se even in their most Se-rich film). The authors responded with: 'Published bandgap bowing relationships for $\text{CdS}_{1-x}\text{Te}_x$ indicate the bandgap reduction of 0.05 eV observed in this work corresponds to a Se content of 10 at% [J. Yang and S.-H. Wei, Chin. Phys. B 28, 086106 (2019)].' However, reviewing the bandgap bowing data from the cited paper Yang paper (Fig 2) reveals that an 0.05 eV Bg reduction corresponds to an ANION alloying fraction change of $x=0.1$ (i.e. 5 at.%, not 10 at% as stated).

The reviewer is of course correct and thanks for picking us up on it. This error does not however appear anywhere in the main text or supporting information requiring modification.

Reviewer #2 (Remarks to the Author):

Revised manuscript by Bowman et al is significantly improved. It is a technically correct study, with interesting data on CdSe(0.1)Te(0.9) photoluminescence and other properties.

Thank you. The reviewer's prior comments and suggestions were very helpful in improving the manuscript.

I will reiterate that results are not relevant for CdSeTe solar cells (please see below). If sentences 1-2 from the abstract and PV-related paragraphs from introduction are removed, results can be published, they will add to the literature on CdSe(0.1)Te(0.9) electronic properties.

We had perhaps misinterpreted this initially and appreciate the reviewer's point on the abstract, that the first two sentences could be misconstrued, and we have removed them. The abstract now reads as follows where we have move away from PV related discussion except where it relates to prior work :

“Evidence from cross-sectional electron microscopy has previously shown that Se passivates defects in CdSe_xTe_{1-x} solar cells, and that this is the reason for better lifetimes and voltages in these devices. Here, we utilise spatially resolved photoluminescence measurements of CdSe_xTe_{1-x} thin films on glass to directly study the effects of Se on carrier recombination in the material, isolated from the impact of conductive interfaces and without the need to prepare cross-sections through the samples. We find further evidence to support Se passivation of grain boundaries, but also identify an increase in below-bandgap photoluminescence that indicates the presence of Se-enhanced defects in grain interiors. Our results show that whilst Se treatment, in tandem with Cl passivation, does increase radiative efficiencies in CdSe_xTe_{1-x}, it simultaneously increases the defect content within the grain interiors. This suggests that although it is overall beneficial, Se incorporation will still limit the maximum attainable optoelectronic properties of CdSe_xTe_{1-x} thin films.”

For the introduction, while we understand the concern we do feel the PV-related paragraphs are needed to put the work into PV context which is the overall motivation of the work.

Manuscript by Bowman is limited to $x < 0.1$ in CdSe(x)Te(1- x). But it has been shown that passivation effect for $x=0.1$ composition is insignificant. E.g., Amarasinghe et al (APL 118, 211102 (2021)) reported TRPL lifetimes as a function of x . These lifetimes were 30 ns for $x=0$, 60 ns for $x=0.1$, 800 ns for $x=0.2$, and 1300 ns for $x=0.3$. Thus, to understand passivation (stated in the title), higher x values must be studied.

For the Se-content we were forced to balance the level of Se content achievable given the device structure. Our entire premise for this work was based on the use of glass substrates to remove the influence of interfaces and provide as “clean” a PL analysis as was possible. Because of the removal of the textured underlayers this modifies the adhesion during Cl treatment and ultimately limits the amount of Se incorporation possible. Whilst ideally we would have liked to study higher Se content, as the reviewer suggests, we were forced to make a choice between that and removal of the interface layers, choosing the latter. We would note that our results are fully consistent with those from Amarasinghe et al., and in reference to their work we do not believe that a doubling of radiative lifetimes is an insignificant result (noting this corresponds to the 0.1 Se content we measure). For $\text{CdSe}_x\text{Te}_{1-x}$ solar cells the quantity of Se typically varies from the front to the back of the thin film, so it is important that the optoelectronic properties are understood at both high and low quantities of Se to optimise full devices. We also expect that the emissive states we have identified will also be present, likely with greater prevalence, in higher content Se content films. However, we do appreciate the reviewer’s concern with regards passivation in the title, so it now reads:

“Spatially resolved photoluminescence analysis of the role of Se in $\text{CdSe}_x\text{Te}_{1-x}$ thin films”

We have also added the following to the main text:

“Whilst the bandgap of the 200 nm CdSe sample indicates a lower Se content than typically found in record-efficiency devices [Green63], the trend towards increased sub-bandgap emission for higher Se content in this study suggests that this emissive state is likely to be prevalent in material with larger Se content. This is especially important in $\text{CdSe}_x\text{Te}_{1-x}$ solar cells where the ratio of Se to Te is graded across the cell.”

[Green63] M. A. Green, E. D. Dunlop, M. Yoshita, N. Kopidakis, K. Bothe, G. Siefer, and X. Hao, Solar cell efficiency tables (version 62), Progress in Photovoltaics: Research and Applications, 31(7), (2023).

“We note that our current fabrication approach was not able to access samples with higher CdSe thickness due to additional delamination problems.”

Preparation of $\text{CdSe}_x\text{Te}_{1-x}$ films with variation in x is widely used, it is a standard approach in this field. E.g., very recent paper (Frouin et al, APL Mater. 12, 031135 (2024)) has reported CL data for $x=0, 0.1, 0.2, 0.3,$ and 0.4 . Frouin et al show that radiative emission changes very substantially for higher x values (Figure 1d and SI). Spatially resolved CL data in Frouin et al is in good agreement with macroscopic PL data in [Kuciauskas2023], measured on samples fabricated at different institutions (NREL for Frouin and Colorado State for Kuciauskas), meaning that defect features in CdSeTe

reproducible for absorbers from different labs. PV relevant compositions must be studied if the goal is to provide guidance to PV material development.

We agree the work mentioned is worth highlighting and we have cited the described work as detailed in response to a comment below. As discussed above we have updated our abstract to focus more on materials than just solar cells but we again emphasise that fabrication on glass is non-trivial in comparison to on electronic transport layers, which restricts the level of Se incorporation.

Second (smaller) point is about excitation conditions used in confocal PL: 5.9×10^6 mW/cm², or approximately 60,000 Suns. Solar cells are used at 1 Sun. Authors show that trapping regime still applies at 60,000 Suns, meaning that material studied by Bowman has a lot of defects. But it is not shown that defects are the same at 60,000 Suns and 1 Sun (e.g., it is likely that some trap states are filled at 60,000 Suns). Using high excitation in confocal PL is understandable, and data is good, but results don't directly apply at 1 Sun. E.g., it might be possible to measure macroscopic PL at 1 Sun, to compare confocal and standard PL emission data.

Yes this would indeed be a logical approach but ultimately our measurement approach came down to the idealised case vs what is experimentally possible. It was essentially a signal to noise issue.

Our initial measurement approach was to study samples using wide-area illumination conditions. However, we found luminescence signals, which did not correspond to the CdTe/Cd_xTe_{1-x} luminescence, which orders of magnitude stronger than the CdTe/Cd_xTe_{1-x} luminescence and thus dominated the signal. These were determined to originate from glass luminescence leaking through pinholes in delaminated regions (determined by comparative measurement of bare substrate illuminated by a 488 nm beam at 5 mW, over an area of 0.006 cm²). This same luminescence dominated macroscopic signals including at 1 sun. It was this difficulty that pushed us towards measurements using a more focused laser beam, which enabled us to excite only a small region of the material and thereby filter out the background noise. We can also comment though that as the majority of trap states in this material are dark traps, they will not influence the shape of the luminescence peak but only the scaling of this peak with laser power (we have covered this in our previous work included as reference [1]), hence the higher illumination intensity is not problematic. We have added the following to the main text for clarification:

“We selected the incident laser intensity (5.9×10^6 mWcm⁻²) to be high enough to allow for efficient mapping, but still within a trap-dominated regime (see Supporting Information Note 6 for further discussion and evidence that the luminescence shape does not change with incident intensity).”

and the following to the methods:

“We also note that we initially attempted PL measurements using wide-field illumination and hyperspectral mapping. In this hyperspectral system we found that signals from the material were overtaken by intrusive luminescence signals from the glass substrate. This initiated our approach of using a Raman Microscope with a focused laser beam which allowed clear resolution of the material luminescence signal.”

[1] A. R. Bowman, S. Macpherson, A. Abfalterer, K. Frohna, S. Nagane, and S. D. Stranks, *Extracting Decay-Rate Ratios From Photoluminescence Quantum Efficiency Measurements in Optoelectronic Semiconductors*, *Phys. Rev. Applied* 17, 044026 (2022).

In summary, a better, more comprehensive paper on this topic was recently published. Bowman et al manuscript can be published as well, as a confirmation of radiative emission mapping with a complimentary method (confocal PL vs CL), but on a more limited range of CdSeTe compositions.

The recent work by Frouin et al., is indeed excellent and highly complementary to this work which we have now cited as follows:

“As already noted, similar effects have been observed in bulk photoluminescence measurements by Kuciauskas et al. and Hu et al. [Kuciauskas2023, Hu2023], and cathodoluminescence measurements by Frouin et al. [Frouin2024], but this is the first photoluminescence evidence of a clear correlation between a controlled Se content and the strength of the defect luminescence.”

“For this defect to be appropriately passivated, its location within the sample must first be understood (going beyond the cathodoluminescence analyses by Frouin et al. [Frouin2024]).”

Additionally, we note that whilst cathodoluminescence and photoluminescence are complementary techniques, the quantitative luminescence efficiencies given by our measurements are a key parameter that cannot be directly accessed through cathodoluminescence. Furthermore, photoluminescence (including at the microscale) is significantly more readily accessible to many laboratories, meaning we hope our results will help inspire future similar analyses.

[Frouin2024] B. Frouin, T. Bidaud, S. Pirotta, T. Ablekim, J. Moseley, W. K. Metzger, and S. Collin, *Quantitative Assessment of Selenium Diffusion and Passivation in CdSeTe Solar Cells Probed by Spatially Resolved Cathodoluminescence*, *APL Materials* 12, 031135 (2024).

[Hu2023] G. Hu, H. Cao, P. Tang, X. Hao, B.-H. Li, H. Li, D. Zhao, W. Li, L. Wu and J. Zhang. *Ultrafast photocarrier dynamics of CdSexTe1-x polycrystalline films under low illumination*. *Solar Energy*. 256, 289-293 (2023)

[Kuciauskas2023] D. Kuciauskas, M. Nardone, A. Bothwell, D. Albin, C. Reich, C. Lee, E. Colegrove. Why Increased CdSeTe Charge Carrier Lifetimes and Radiative Efficiencies did not Result in Voltage Boost for CdTe Solar Cells. *Adv. Energy Mater.* 13, 2301784 (2023)

Reviewer #3 (Remarks to the Author):

In this manuscript, the author aims to understand the influence of Se introduction on the properties of CdSexTe1-x films, which was formed by sequentially depositing CdSe and CdTe films and followed by high-temperature heat treatment in a chlorine containing atmosphere, through micro PLQY characterization technology and other characterization methods. As stated in the previous review comments, the research perspective proposed in the manuscript on this scientific issue has not been reported in relevant literature. It is expected that researchers in the field of CdTe solar cells will be very interested in the research results of this manuscript. So, this paper is well worth publishing.

We thank the reviewer for the supportive comments and for their previous comments, which have helped improve the paper.

However, a very important factor to consider when conducting this study is to clarify which scale range of CdSexTe1-x films correspond to the measured PLQY results, and whether the specific Se content and structure of CdSexTe1-x films in this scale range are clear. Therefore, in the previous review comments, it was suggested that the author conduct SIMS characterization to clarify the distribution of Se in CdSexTe1-x thin films. From the SIMS characterization results submitted by the author, it can be seen that the concentration of Se in the direction from the glass substrate to the air side gradually decreases.

This is consistent with the process of preparing the sample mentioned in the manuscript, which involves depositing a CdSe layer first, followed by depositing a CdTe layer, allowing Se to diffuse during the high-temperature process to obtain a specific CdSexTe1-x thin film sample. Using this process, it is possible to obtain CdSexTe1-x samples with uniformly distributed Se in the direction perpendicular to the substrate surface, but very harsh conditions may be required, such as very long time, appropriate high temperature, etc.

Obviously, the SIMS characterization results provided in the revised manuscript indicate that the process used by the author to prepare the samples did not ensure the uniform longitudinal distribution of Se content in CdSexTe1-x thin film samples.

The reviewer is correct, any additional Se-diffusion requires harsher processing conditions that the sample structure could not withstand. As mentioned in response to reviewer 2 we were forced to make an experimental choice between higher Se diffusion or removing the influence of the interface layers. We chose the latter as this was deemed to give us clearer data, albeit at lower than device-optimal Se content, but as noted the findings and mechanisms should still be applicable at higher Se-content.

Diffusion from a CdSe layer is the most common mechanism to form CST in the literature and a degree of Se grading is widely observed and is typically the targeted structure. There are few, if any, reports of uniformly distributed Se/Te within a $\text{CdSe}_x\text{Te}_{1-x}$ thin film via this fabrication method, primarily due to the harsh conditions required (as the reviewer highlights). An example can be seen in the data below from Li et al for a 20% efficient device where there is significant Se grading.

Li, D. B. et al. 20%-efficient polycrystalline Cd(Se,Te) thin-film solar cells with compositional gradient near the front junction. Nat Commun 13, (2022).

We do agree this is worth clarifying though so have added the following to the text:

“Whilst the bandgap of the 200 nm CdSe sample indicates a lower Se content than typically found in record-efficiency devices [Green63], the trend towards increased sub-bandgap emission for higher Se content in this study suggests that this emissive state is likely to be prevalent in material with larger Se content. This is especially important in $\text{CdSe}_x\text{Te}_{1-x}$ solar cells where the ratio of Se to Te is graded across the cell.”

[Green63] M. A. Green, E. D. Dunlop, M. Yoshita, N. Kopidakis, K. Bothe, G. Siefer, and X. Hao, Solar cell efficiency tables (version 62), Progress in Photovoltaics: Research and Applications, 31(7), (2023).

“We note that our current fabrication approach was not able to access samples with higher CdSe thickness due to additional delamination problems.”

Furthermore, the authors clearly indicate in the sample testing section on page 12 of the revised manuscript that when performing PLQY testing, using 488nm excitation light, the absorption depth is only ~ 88 nm. In the sample preparation section on page 13, it is also clearly stated that the sample thickness is $2\sim 2.5$ microns. According to literature, using a process similar to the one described in the manuscript, in which the samples were treated in a chlorine-containing atmosphere, the diffusion depth of Se in CdTe is usually ~ 1 micron [1, 2]. Combined with the SIMS characterization results provided by the manuscript, which reflects the distribution of Se elements perpendicular to the substrate direction, the PLQY value only comes from the contribution of less than 100nm depth range near the air side in CdSexTe1-x thin film samples, which is the region with the lowest Se content in CdSexTe1-x thin film samples. So, based on the current PLQY measurements in the revised manuscript and the characterization results of Se content in CdSexTe1-x thin film samples, it is scientifically insufficient and logically not rigorous to derive the rule of Se content's influence on the properties of CdSexTe1-x thin films.

There is a slight misunderstanding here: as our measurement is steady-state photoluminescence, this means that the charges diffuse prior to luminescence. As the reviewer correctly states the charge diffusion length is ~ 1 μm in these samples, meaning that the photoluminescence signal originates from this ~ 1 μm region rather than the 88 nm suggested. When referring to our SIMS results, in this 1 μm region there is less than a 20 % variation in the Se/Te ratio for all regions studied, which for any study on CdSexTe1-x films is acceptable uniformity. We exemplify this in S1c below, which highlights the region the photoluminescence originates from.

with Supporting Information Figure S1:

Figure S1. SIMS intensity for a) Se and b) Te as a function of Etch number (with 0 etches being the air-exposed surface). c) ratio of Se to Te intensities, with the region that the photoluminescence (PL) originates from marked. Legend in c) applies to all plots.”

and accompanying text:

“Finally, in Figure S1c we present the approximate region of the sample which the photoluminescence originates from, following discussion in the main text.”

We have also updated the main text to read:

“Therefore, as the half of our films closer to air have a uniform Se to Te ratio (Supporting Information Note 2), our photoluminescence signals originate from a region where the ratio of Se to Te can be considered to be constant. We plot this explicitly in Supporting Information Figure S1.”

Finally, we have updated the manuscript so we now refer to ‘air side’ or ‘substrate side’ of the film, rather than ‘front side’ and ‘back side’.

In summary, the following suggestions are proposed:

(1) Prepare CdSexTe1-x thin films with uniform longitudinal distribution of Se content using other techniques, and then perform PLQY characterization.

Hopefully we have adequately addressed this above, but we reemphasise that our PLQY results originate from a region with a highly uniform Se/Te ratio (within 20 %).

(2) Alternatively, a technique similar to Ar ion etching in XPS can be used to sequentially etch CdSexTe1-x thin film samples, reducing their thickness and characterizing PLQY, in order to obtain the variation pattern of how the longitudinal Se content distribution affects the PLQY measurement results of the samples.

We again note that steady state photoluminescence measurements are not a surface sensitive measurement but are always a convolution of charge diffusion throughout the film, meaning that any etching processes would explore different (approximately 1 μm thick) regions of the sample. As our study already measures the 1 μm region with the most uniform Se/Te ratio: we do not believe that additional insights would be obtained by etching to explore less uniform sample regions. Furthermore, most etching techniques can significantly alter exposed surfaces, resulting in different passivation and surface termination, which would in turn affect photoluminescence results. This is however an excellent idea for a future complementary study which we may well undertake, perhaps with the addition of some HAXPES measurements if we can secure beamtime.

(3) To increase the absorption depth of excitation light, an excitation light source with photon energy slightly higher than the bandgap width of CdSexTe1-x can be used in PLQY testing. But this not only requires a uniform longitudinal distribution of Se content in CdSexTe1-x films, but also requires that the thickness of CdSexTe1-x films is equivalent to the absorption depth, ensuring that the results of PLQY can reflect the chemical environment related to Se in the overall CdSexTe1-x films.

While we appreciate the reviewer's suggestion, exciting close to the bandgap of the sample results in significant additional complications when calculating photoluminescence quantum yield. Specifically, it is difficult to disentangle the laser signal from a luminescence signal in the same wavelength region given that one is six orders of magnitude weaker. Consequently, it is common in photoluminescence studies of thin films to use an excitation wavelength significantly higher than the luminescence peak, to avoid any difficulty in deconvoluting signals (e.g. Frohna et al. 2022). Our measurements have studied a region with uniform Te/Se ratio and charge diffusion means we are studying a region approximately 1 μm thick, meaning we have already achieved what the reviewer is suggesting here.

[Frohna2022] K. Frohna, M. Anaya, S. Macpherson, et al. Nanoscale chemical heterogeneity dominates the optoelectronic response of alloyed perovskite solar cells. *Nat. Nanotechnol.* 17, 190–196 (2022). <https://doi.org/10.1038/s41565-021-01019-7>

REVIEWERS' COMMENTS

Reviewer #1 (Remarks to the Author):

Thank you for addressing the concerns.

Reviewer #3 (Remarks to the Author):

The author fully answered the review comments and made necessary amendments to the manuscript. So now the manuscript can be accepted for publication.